# BioGAN: Enhancing Transcriptomic Data Generation with Biological Knowledge

**DOI:** 10.3390/bioengineering12060658

**Published:** 2025-06-16

**Authors:** Francesca Pia Panaccione, Sofia Mongardi, Marco Masseroli, Pietro Pinoli

**Affiliations:** Department of Electronics, Information, and Bioengineering, Politecnico di Milano, 20133 Milan, Italysofia.mongardi@polimi.it (S.M.);

**Keywords:** synthetic transcriptomic data, generative adversarial networks, graph neural networks, generative artificial intelligence, biologically informed methods, computational biology

## Abstract

The advancement of computational genomics has significantly enhanced the use of data-driven solutions in disease prediction and precision medicine. Yet, challenges such as data scarcity, privacy constraints, and biases persist. Synthetic data generation offers a promising solution to these issues. However, existing approaches based on generative artificial intelligence often fail to incorporate biological knowledge, limiting the realism and utility of generated samples. In this work, we present BioGAN, a novel generative framework that, for the first time, incorporates graph neural networks into a generative adversarial network architecture for transcriptomic data generation. By leveraging gene regulatory and co-expression networks, our model preserves biological properties in the generated transcriptomic profiles. We validate its effectiveness on *E. coli* and human gene expression datasets through extensive experiments using unsupervised and supervised evaluation metrics. The results demonstrate that incorporating a priori biological knowledge is an effective strategy for enhancing both the quality and utility of synthetic transcriptomic data. On human data, BioGAN achieves a 4.3% improvement in precision and an up to 2.6% higher correlation with real profiles compared to state-of-the-art models. In downstream disease and tissue classification tasks, our synthetic data improves prediction performance by an average of 5.7%. Results on *E. coli* further confirm BioGAN’s robustness, showing consistently strong recall and predictive utility.

## 1. Introduction

Since the completion of the Human Genome Project in the early 2000s, genomics has increasingly become a discipline with strongly data-driven components. Nowadays, artificial intelligence (AI)-based solutions applied to genomic data have shown wide applicability in early and precise disease diagnosis, improved outcome prediction, and personalized therapy selection. This progress has greatly benefited from high-throughput sequencing technologies, which enable the large-scale sequencing of RNA and DNA sequences. For example, next-generation sequencing (NGS) is widely used in genomics and transcriptomics. These technologies allow researchers to collect vast amounts of data from a single experiment, thus enabling the comprehensive analysis of biological systems. Transcriptomics, for example, uses modern technology to measure gene expression across thousands of genes, offering a holistic view of the transcriptome–the complete set of RNA transcripts produced by the genome—capturing the dynamic and context-dependent activity of genes in different tissues, developmental stages, or disease conditions. This enables the identification of differentially expressed genes, revealing active biological pathways, uncovering regulatory mechanisms such as alternative splicing, and providing insights into cell-type composition and disease-specific expression patterns.

In this framework, access to large, high-quality transcriptomic datasets, encompassing a wide range of diseases and phenotypic conditions, is paramount to advancing the capabilities of AI tools and producing trustworthy, fair, and robust methods [1,2,3].

However, collecting genomic data presents several intrinsic challenges. Firstly, the high cost of acquisition makes it difficult to compile large datasets, particularly in clinical settings. Secondly, human genomics is highly heterogeneous, making it challenging to collect datasets where no subgroup is underrepresented. Thirdly, different laboratories often collect and process data using slightly different protocols, leading to technical biases and inconsistencies. Lastly, genomic data are highly sensitive, and strict regulations (e.g., the General Data Protection Regulation [4] and the Health Insurance Portability and Accountability Act [5]) impose significant limitations on their collection and distribution.

Deep generative models can generate synthetic samples that faithfully reflect the statistical properties of real data, offering a promising solution to address scarcity, biases, imbalance, and privacy constraints. Although unbiased generators can produce relatively high-quality data, in many fields, incorporating domain knowledge into the generation process has been shown to increase the realism of synthetic data. However, unlike other scientific fields, such as physics, this approach remains largely unexplored in genomics due to the complexity of biological systems, gaps in our understanding of molecular mechanisms, and strong variability among samples.

Moving in this direction, we propose BioGAN, an original deep generative model based on a Wasserstein Generative Adversarial Network with Gradient Penalty [6,7] designed for transcriptomic data synthesis. BioGAN leverages a graph neural network (GNN) that operates on an input graph representing known biological properties (e.g., regulatory relationships) to guide the generation of novel samples in a more realistic and biologically plausible manner. The use of graph-based architectures is particularly well suited to transcriptomic data, where gene expression is shaped by complex, non-linear dependencies among genes. GNNs are inherently capable of modeling these relationships, and they have already demonstrated strong performance in a range of biomedical applications, including disease gene prioritization, the prediction of drug response, and the inference of a cell-type-specific regulatory network. However, this strategy has not yet been explored for generative modeling in the omics domain, marking a novel and promising direction for enhancing both the biological fidelity and practical utility of synthetic transcriptomic data.

A further challenge in generating synthetic data lies in the validation procedure and the metrics used to assess the quality of the generated data. This is particularly hard in high-dimensional spaces such as transcriptomics, where gene expression profiles contain tens of thousands of features. Current evaluation metrics for synthetic data (e.g., precision, recall, correlation, and Wasserstein distance) often produce conflicting results, complicating the assessment of data realism.

Our manuscript addresses these challenges, and its main contributions can be summarized as follows:Graph-informed generative modeling for omics data: This study introduces a novel framework that integrates graph neural networks (GNNs) into deep generative models to improve the biological fidelity of synthetic omics data.Comprehensive evaluation of synthetic data: The realism and utility of the generated data are assessed through a multi-faceted validation framework, including the following: (a) supervised machine learning models for classification tasks; (b) unsupervised statistical comparisons between real and synthetic data; and (c) feature-level fidelity assessments to quantify biological consistency.

The code is available at https://github.com/DEIB-GECO/BioGAN (accessed on 8 June 2025).

## 2. Background

This section provides a detailed overview of generative models, graph theory, and biological networks. It highlights important architectures and methods, particularly focusing on how biological graphs can be used to represent interactions and dependencies in biological systems.

### 2.1. Introduction to Generative Models

Generative models are a class of machine learning models that aim to approximate probability distributions over high-dimensional data. Given a dataset $X=\left\{x_{1},x_{2},\ldots ,x_{n}\right\}\subseteq R^{k}$ of *n* samples in *k*-dimensional space, the goal is to learn a distribution $P_{\mathrm{data}}\left(X\right)$ that captures the underlying structure of the data points in a high-dimensional space. Deep learning generative models, such as variational autoencoders (VAEs) [8] and generative adversarial networks (GANs) [9], perform well at this task.

#### 2.1.1. Variational Autoencoders

The VAE framework introduces a probabilistic model to learn a representation of the input data in a latent space $R^{d}$, typically with d≪k. A VAE is composed of two networks: an encoder, which maps each data point $x\in R^{k}$ to its latent representation $z\in R^{d}$, and a decoder, which generates a reconstruction, $\hat{x}$, from the latent representation. The encoder approximates the posterior distribution $q_{\phi }\left(z|x\right)$, and the decoder models the likelihood of the data, given the latent representation, $p_{\theta }\left(x|z\right)$. The VAE objective function, which combines a reconstruction term and a regularization term, is given by the evidence lower bound (ELBO), expressed as follows:$$L_{\theta ,\phi }\left(x\right)=E_{q_{\phi }\left(z|x\right)}\left[\log p_{\theta }\left(x|z\right)\right]-\mathrm{KL}\left(q_{\phi }\left(z|x\right)\;∥\;p_{z}\left(z\right)\right).$$
where the first term represents the reconstruction error (with E denoting the expected value under the approximate posterior $q_{\phi }\left(z|x\right)$), and the second term is the Kullback–Leibler (KL) divergence between the approximate posterior and the prior distribution $p_{z}\left(z\right)$. By maximizing the ELBO, the model learns to generate data by sampling from the latent space and reconstructing the original input.

#### 2.1.2. Generative Adversarial Networks

GANs are a class of generative models that consist of two networks, namely a generator and a discriminator, that are trained in a game-theoretic framework to learn the distribution of data. The generator takes a latent vector $z∼p_{z}$ (typically a multivariate Gaussian), where ∼ denotes sampling from a distribution, and produces synthetic data, G(z), while the discriminator attempts to distinguish between real data, $x∼p_{\mathrm{data}}$, and generated samples. The objective is as follows:$$\min_{G}\max_{D}E_{x∼p_{\mathrm{data}}}\left[\log D\left(x\right)\right]+E_{z∼p_{z}}\left[\log\left(1-D\left(G\left(z\right)\right)\right)\right].$$
However, one common issue with training GANs is the instability of the training process, often due to the vanishing gradient problem when the discriminator becomes too good at distinguishing real from fake data. The Wasserstein GAN (WGAN) [6] addresses this issue by replacing the standard loss with the Wasserstein distance, leading to more stable gradients. The WGAN objective is as follows:$$\min_{G}\max_{D}E_{x∼p_{\mathrm{data}}}\left[D\left(x\right)\right]-E_{z∼p_{z}}\left[D\left(G\left(z\right)\right)\right]$$
where D(x) and G(z) are the outputs of the discriminator (critic) and generator, respectively. The discriminator is trained to approximate the Wasserstein distance between the distributions of real and generated data. To enforce the 1-Lipschitz constraint on the discriminator and avoid gradient explosion—thus stabilizing training—a gradient penalty term is introduced in the objective, leading to the Wasserstein GAN with Gradient Penalty (WGAN-GP) formulation:$$L_{\mathrm{WGAN}-\mathrm{GP}}=E_{x∼p_{\mathrm{data}}}\left[D\left(x\right)\right]-E_{z∼p_{z}}\left[D\left(G\left(z\right)\right)\right]+\lambda E_{\hat{x}}[(∥\nabla_{\hat{x}}D\left(\hat{x}\right){∥}_{2}-1{)}^{2}],$$
where $\hat{x}$ is sampled between real and generated samples, and λ controls the gradient penalty strength. Extending this framework, Conditional GANs (cGANs) [9] incorporate auxiliary information *c* (e.g., class labels, attributes, or other metadata) into both the generator and the discriminator, enabling the generation of data samples conditioned on specific characteristics. The objective function of a cGAN is as follows:$$\min_{G}\max_{D}E_{x∼p_{\mathrm{data}}}\left[\log D\left(x|c\right)\right]-E_{z∼p_{z}}\left[\log\left(1-D\left(G\left(z|c\right)\right)\right)\right].$$
In this formulation, the generator, *G*, generates data conditioned on both the latent variable, z, and the condition, c, while the discriminator, *D*, classifies whether a sample, x, is real or fake, given the condition c.

### 2.2. Foundations of Graphs and Graph Neural Networks

A graph, G=〈V,E〉, consists of a set nodes (vertices), *V*, and a set of edges, *E*. Given two nodes, i,j∈V, and edge, (i,j)∈E, means that node *i* is connected to node *j*. Furthermore, edges can be directed and weighted.

Graphs provide a robust framework for modeling complex relationships and dependencies in structured data, and graph neural networks (GNNs) are designed to learn from such data. In GNNs, for each node, *u*, a feature vector, $x_{u}\in R^{k}$, is computed, and the feature matrix X for all nodes is as follows:$$X={\left(\begin{array}{cccc} x_{1} & x_{2} & \ldots & x_{\left|V\right|} \end{array}\right)}^{⊤}$$
GNNs compute the values of each $x_{i}$ by message-passing the mechanism through which nodes aggregate information from their neighbors, refining their feature vectors through multiple iterations. At each iteration *T*, the feature vector of the node *i* is computed on the previous values of the nodes in the graphs as follows:$$x_{i}^{T}=\phi (x_{i}^{T-1}),\underset{j\in N(i)}{⨁}(\psi \left(x_{i}^{T-1}),x_{j}^{T-1},e_{ij}\right)$$
where ϕ and ψ are generic functions, N(i)=j∈V:(i,j)∈E denotes the set of neighbors of *i*, ⊕ is a permutation-invariant aggregation function (e.g., mean or sum), and $e_{ij}$ represents the features of the connection between the nodes *i* and *j* (e.g., the weight). Different updated functions (also called *layers*) exist to capture different aspects of the graph. For example, some layers aggregate information from direct neighbors (1-hop), while others consider multiple hops (2-hop or more).

GNNs typically assume homophily, meaning similar nodes tend to be connected, as captured via the edge homophily ratio:$$H\left(G\right)=\frac{1}{\left|E\right|}\sum_{(j,k)\in E}\left(\mathrm{similarity}\left(j,k\right)\right),$$
where *similarity* is a domain-dependent function. High homophily indicates that the connected nodes share similar features. Usually, GNNs struggle to generalize in the presence of low homophily.

#### Biological Graphs

Given the complexity of biological systems, and in particular the relationships among their components, graphs offer a convenient means of representation. Gene Co-Expression Networks (GCN) and Gene Regulatory Networks (GRN) have been widely adopted to capture different types of interactions derived from transcriptomic data [10,11].

In Gene Co-Expression Networks (GCNs), genes correspond to nodes, and edges indicate significant correlations among gene expression levels. These undirected graphs reflect coordinated expression patterns and can vary across conditions such as tissue types or disease states, revealing tissue-specific or context-specific gene modules. Conversely, GRNs are typically modeled as directed graphs, where nodes represent genes and edges indicate regulatory influences, often mediated via transcription factors binding to promoter regions or enhancers. These interactions may be inferred through experimental techniques (e.g., ChIP-seq) or computational methods (e.g., mutual information and Bayesian inference) [12]. Edges are often annotated to reflect activation or repression (e.g., +1/−1), enabling GRNs to represent causal or mechanistic regulatory relationships.

### 2.3. Related Works

Recent studies have explored the use of deep learning models, such as VAEs [8] and GANs [9], to generate omics data.

Viñas et al. [13] introduced a conditional WGAN-GP [7] model to generate realistic transcriptomics data for *Escherichia coli* and humans. A key feature of their model is the use of word embeddings to model categorical covariates during the generation process. For example, tissue type can be represented as a categorical variable, which is transformed into a one-hot vector and then mapped to a continuous embedding vector. This approach improves upon existing simulators like SynTReN [14] and GeneNetWeaver [15], which struggle to preserve important gene expression relationships, such as transcription factor–target gene (TF–TG) and target gene–target gene (TG–TG) interactions. However, their model does not incorporate prior biological knowledge, and this limits its ability to model more complex biological relationships.

Lacan et al. [16] proposed an attention-based GAN model for transcriptomics data augmentation. Their model uses an attention mechanism to focus on gene pairs that are known to have strong co-expression or protein–protein interactions (PPIs). However, their findings showed that the model’s performance did not improve significantly when the attention mask was altered or randomly generated. This suggests that the performance gains may be more due to the complex architecture of the model, rather than the integration of domain-specific biological knowledge.

A more robust approach to incorporating prior knowledge into generative models is demonstrated by Liu et al. [17] with the Generative Modelling with Graph Learning (GOGGLE) framework. This framework combines a VAE with a GNN to generate data while capturing the relational structure between features. The relationships between features are represented as a sparse graph, where the nodes correspond to variables and edges encode their dependencies. The model uses this graph structure to guide the generation process, ensuring that only relevant relationships influence the synthetic data. By integrating prior knowledge through predefined adjacency matrices and a regularization term in the loss function, GOGGLE can generate data that better reflects known relationships. However, a limitation of this approach is its computational cost, which makes it feasible only for datasets with fewer than 100 features.

In addition, when working with graph-structured data, especially in cases where heterophily is present (i.e., connected nodes differ significantly in their features), traditional GNNs may struggle. To address this, Zhu et al. [18] proposed several strategies to improve GNNs in heterophilic settings. These strategies include separating node embeddings (ego embeddings) from neighbor embeddings to capture differences between nodes more effectively. They also suggested explicitly aggregating information from higher-order neighborhoods during the message-passing process and using a residual mechanism to combine intermediate node representations. These strategies were implemented in the H2GCN [19] model, which outperforms conventional convolutional layers in both homophilic (nodes are similar) and heterophilic (nodes are different) graph scenarios, improving the model’s ability to generalize.

## 3. Materials and Methods

This section outlines the architecture of BioGAN, which leverages prior biological knowledge in the form of graphs. It also describes the datasets, data pre-processing steps, and biological insights integrated into the model.

### 3.1. BioGAN Architecture

Our proposed model, called Graph WGAN-GP, synthesizes gene expression data by integrating biological knowledge into the generation process. This is achieved through a graph-based generator built with GNN layers, which leverages the connections between genes within biological networks as prior information.

#### 3.1.1. Graph-Based Generator

BioGAN implements an original graph-based generator for the in silico synthesis of samples. The data generation process starts with a noise vector $z\in R^{d}$, drawn from a standard normal distribution. The noise vector is concatenated to the embedded covariates $c\in R^{m}$ to create the input vector, v=[z,c]. A dense layer processes v to assign an initial scalar value to each node:$$h^{0}=\sigma \left(W_{I}v+b_{I}\right),$$
where $h^{0}$ is a uni-dimensional vector representing, $W_{I}\in R^{p\times (d+m)}$ is a learnable weight matrix, $b_{I}$ is a bias term, and σ is a non-linear activation function. The dimension of $h^{0}$ is set to match the number of genes in the dataset, ensuring that each element of the vector serves as the initial embedding of each node in the graph. This initial embedding is refined through a message-passing framework that iteratively updates node features by aggregating information from neighboring nodes via a weighted adjacency matrix, A. The matrix $A^{\left|V|\times |V\right|}$, with |V| equal to the number of nodes (genes), is computed differently according to the scenario and the type of available information:For a GCN, A is undirected, weighted, symmetric, capturing the level of similarity between gene expressions. For each pair of genes $g_{i}$ and $g_{j}$, $A_{i,j}=A_{j,i}=\mathrm{corr}\left(g_{i},g_{j}\right)$. The function *corr* usually corresponds to the Pearson correlation. Several variants may be applied: for example, elements of A, whose absolute value is below a certain threshold, are set to zero to avoid spurious correlations. In other cases, A is made binary, preserving only pairs of genes displaying strong correlation (or anti-correlation).**Example:** Suppose genes $g_{1}$ and $g_{2}$ show a Pearson correlation of 0.92 across all liver samples. Then, $A_{g_{1},g_{2}}=A_{g_{2},g_{1}}=0.92$, or 1 if binarized. If gene $g_{3}$ has a correlation of 0.3 with $g_{1}$, and the threshold is 0.8, then $A_{g_{1},g_{3}}=A_{g_{3},g_{1}}=0$.For GRN A, is directed and unweighted, capturing which genes regulate the expression of the other. Specifically, $A_{i,j}=1$ if $g_{i}$ positively regulates $g_{j}$, $A_{i,j}=-1$ if $g_{i}$ negatively regulates $g_{j}$, and $A_{i,j}=0$ otherwise.**Example**: If gene $g_{1}$ is known to activate gene $g_{2}$, and repress gene $g_{3}$, then $A_{g_{1},g_{2}}=1$, $A_{g_{1},g_{3}}=-1$; notice that regulation is not necessarily reciprocal.

Within the BioGAN, for the message-passing layers, we implemented two options, namely GraphConv [20] and H2GCN [19]. In particular, H2GCN is a valid option for heterophilic graphs. The GraphConv layer aggregates features from direct (1-hop) neighbors, aligning well with homophilic graphs where similar nodes are often connected. In each GraphConv layer *t*, a transformation is applied to the feature vector of each node at the layer t−1 as follows:$$h_{i}^{\left(t\right)}=\sigma \left(h_{i}^{\left(t-1\right)}\cdot W_{1}^{\left(t\right)}+\underset{j\in N(i)}{⨁}A_{j,i}\cdot h_{j}^{\left(t-1\right)}\cdot W_{2}^{\left(t\right)}\right)$$
where $h_{i}^{\left(t\right)}$ is the feature vector of node *i* after the *t* layer, $A_{j,i}$ denotes the edge weight from source node *j* to target node *i*, $W_{1}^{\left(l\right)}$ and $W_{2}^{\left(l\right)}$ are learnable weight matrices, σ denotes a component-wise non-linear function, and ⊕ is a permutation-invariant aggregation function.

On the other hand, the H2GCN layer is designed to address heterophily, a condition frequently observed in biological networks due to hierarchical relationships. The H2GCN layer extends the aggregation process beyond immediate neighbors and distinguishes between self-embeddings and neighbor embeddings, preserving the unique characteristics of each neighborhood. In H2GCN, the update rule for node *v* aggregates both 1-hop and 2-hop neighborhoods separately:$$h_{i}^{\left(l\right)}=\left(\sum_{j\in N_{1}\left(i\right)}\frac{A_{j,i}}{\sqrt{\left|N(i)||N(j)\right|}}h_{j}^{\left(l-1\right)}W_{1}^{\left(l\right)}\right)\;\|\;\left(\sum_{k\in N_{2}\left(i\right)}\frac{A_{k,i}}{\sqrt{\left|N(i)||N(k)\right|}}h_{k}^{\left(l-1\right)}W_{2}^{\left(l\right)}\right)$$
where $N_{1}\left(i\right)$ is the set of 1-hop neighbors of node *i*, $N_{2}\left(i\right)$ is the set of 2-hop neighbors, $W_{1}^{\left(l\right)}$ and $W_{2}^{\left(l\right)}\in R^{p\times p}$ are the learnable weight matrices for the 1-hop and 2-hop neighbor aggregations, respectively, and the operator || refers to the concatenation operation. To obtain the final feature vector for a node, the H2GCN layer concatenates all its intermediate representations ($h_{i}^{\left(j\right)}$) and passes it through an MLP to obtain the final output value $y_{i}$, as follows:$$h_{i}^{\left(final\right)}=(h_{i}^{\left(0\right)}\left|\right|h_{i}^{\left(1\right)}\left||,\ldots ,|\right|h_{i}^{\left(L\right)})$$$$y_{i}=f\left(h_{i}^{\left(final\right)},W_{c}\right)$$
where *f* is any activation function, which can also be omitted, and $W_{c}$ is a learnable weight matrix. Through this adaptable architecture, Graph WGAN-GP effectively combines the structural knowledge embedded in the adjacency matrix with flexible aggregation strategies (GraphConv and H2GCN layers, or others). This approach enables the model to account for diverse biological contexts, whether characterized by homophilic or heterophilic relationships.

#### 3.1.2. Overall Model Architecture

Figure 1 illustrates the architecture BioGAN. On the left, the noise vector z∼N(0,1) and covariates are concatenated and passed through a dense layer (blue) to generate the initial feature vector, $h^{0}$. In the generator (center), this vector undergoes multiple rounds of message-passing neural networks (MPNNs), leveraging a known biological graph structure to initialize the graph topology. The final output, $h^{\left(L\right)}$ (yellow), represents the synthesized gene data. The critic (right) consists of an MLP that analyses this output and assigns a score. All the deep learning models that we benchmarked were implemented using Python (version 3.10), PyTorch (version 2.3.1), and PyTorch Geometric (version 2.5.3) for GNNs. Specifically, for the graph convolutional layers, we used GraphConv https://pytorch-geometric.readthedocs.io/en/2.5.3/generated/torch_geometric.nn.conv.GraphConv.html (accessed on 8 June 2025) and the original code of H2GCN https://github.com/GemsLab/H2GCN (accessed on 8 June 2025).

### 3.2. Datasets

To test the effect of incorporating biological knowledge into the generative process, we compared BioGAN with classical unbiased methods on two datasets: a collection of microarray samples for *E. coli* retrieved from the Many Microbe Microarrays Database (M^3*D*^) [21] and a collection of RNA-seq experiments on *H. sapiens* tissues gathered from GTEx [22], a research initiative that collects gene expression and variant data across multiple human tissues.

#### 3.2.1. *E. coli* Microarray Data

The *E. coli* single-channel Affymetrix microarray data from the M^3*D*^ database consists of 907 samples and 7459 probes. We considered only the 2648 probes that are part of at least one regulatory interaction in the RegulonDB database [23]. Gene expression values were min–max-normalized to the interval [0,1]. We considered RegulonDB a source of biological information. It provides comprehensive data on the regulatory networks in *E. coli*, including transcriptional regulation, gene expression, and functional annotation. We constructed a directed graph where nodes represent genes and edges denote regulatory relationships, either activation or inhibition. We included 5775 edges (approximately 0.08% of all possible interactions). Activation interactions were weighted +1, while inhibitory interactions were weighted −1.

#### 3.2.2. *H. sapiens* RNA-Seq Data

As a second dataset, we considered RNA-seq experiments collected from GTEx, comprising 4578 samples of 20 different human tissues (Figure A1 in Appendix A) and a library of 19,075 protein-coding genes. We removed “greedy” samples dominated by a small number of genes, leaving 4381 samples. Log_2_ normalization and standard scaling were applied to the gene expression values.

We used two sources of prior knowledge. We built tissue-specific gene correlation networks, computed directly from the GTEx samples. The correlation matrix was filtered to retain only those values whose absolute value was above a predetermined threshold. We selected two correlation thresholds, 0.8 and 0.9, based on both biological and computational considerations. Biologically, these high thresholds ensure that only the strongest and most meaningful gene–gene associations are retained. Doing so ensures that each gene is influenced primarily by its most relevant neighbors, aligning with the understanding that high correlation values are more likely to represent functionally or regulatory significant relationships in gene co-expression analysis. From a computational perspective, using high thresholds yields sparser graphs, significantly reducing the number of edges. This sparsity results in more efficient training, as the number of GNN operations is directly related to the graph density. A lower threshold would have included a vast number of weak correlations, dramatically increasing the computational burden and potentially making training infeasible at scale. Although one might consider optimizing the correlation threshold as a hyperparameter, we deliberately chose not to do so. This decision was motivated by the prohibitive cost of retraining GNN-based models across many threshold settings on large graphs and by the limited added value of including marginally informative interactions. In practice, lower thresholds introduce many edges that have little influence on the downstream generation process while increasing the risk of overfitting. By fixing biologically and computationally justified thresholds, we ensure that the model remains focused and aligned with the biological structure of the data.

## 4. Computational Experiments

This section outlines the experimental setups; we start by detailing the evaluation metrics considered, and then we describe the experimental protocol for the two datasets and report the results.

### 4.1. Evaluation Metrics

Assessing the quality of synthetic data requires appropriate evaluation metrics, as there is no universal standard due to various application domains. In this work, we employ multiple metrics to comprehensively compare real and synthetic data. The following sections detail each metric used.

#### 4.1.1. Unsupervised Metrics

Unsupervised metrics evaluate model outputs by analyzing intrinsic data properties without relying on ground truth labels. Specifically, we used *precision and recall* and the *correlation coefficient*.

Precision and recall [24] are metrics used to assess the quality and relevance of the generated data with respect to the real data. Given the real dataset $X=\{x_{1},\ldots ,x_{n}\}$ and the generated dataset $\hat{X}=\left\{{\hat{x}}_{1},\ldots ,{\hat{x}}_{m}\right\}$, the first step is to compute the manifolds for the real and generated samples $H_{X}^{t}$ and $H_{\hat{X}}^{t}$, respectively. This is done by forming a sphere around each point of the two datasets with a radius corresponding to the Euclidean distance to *t*-th closest neighbor within the same dataset. A binary function, f(h,H), is then defined to evaluate whether the sample, *h*, falls within the manifold, *H*.

Precision and recall can be computed accordingly:$${\mathrm{Precision}}_{t}\left(X,\hat{X}\right)=\frac{1}{|\hat{X}|}\sum_{\hat{x}\in \hat{X}}f\left(\hat{x},H_{X}^{t}\right),\;{\mathrm{Recall}}_{t}\left(X,\hat{X}\right)=\frac{1}{\left|X\right|}\sum_{x\in X}f\left(x,H_{\hat{X}}^{t}\right).$$

In our experiments, we considered precision and recall along with the 10th nearest neighbors.

The correlation coefficient [16] assesses how well the synthetic data preserve gene–gene relationships. This method, based on the Pearson correlation coefficient, quantifies similarity by comparing pairwise relationships between genes in real and generated data.

Given two symmetric n×n matrices, X and $\hat{X}$, representing the pairwise Pearson correlations between all genes in the real and generated datasets, the correlation coefficient is defined as follows:$$c\left(X,\hat{X}\right)=\sum_{i=1}^{n-1}\sum_{j=i+1}^{n}\frac{\left(X_{i,j}-l\left(Y\right)\right)}{r(X)}\cdot \frac{\left({\hat{X}}_{i,j}-l\left(\hat{X}\right)\right)}{r(\hat{X})}$$
where, for a given square matrix, *A*, l(A) and r(A) denote the mean and standard deviation of the pairwise values in matrix A, respectively, and are defined as follows:$$l\left(Y\right)=\frac{2}{n(n-1)}\sum_{i=1}^{n-1}\sum_{j=i+1}^{n}Y_{i,j},\;r\left(Y\right)=\sqrt{\frac{2}{n(n-1)}\sum_{i=1}^{n-1}\sum_{j=i+1}^{n}{\left(Y_{i,j}-l\left(Y\right)\right)}^{2}}.$$

#### 4.1.2. Supervised Metrics—Detectability and Utility

Supervised metrics evaluate the quality of synthetic data by training classifiers on different combinations of real and synthetic data and comparing their performance.

The detectability measure is based on the ability of a binary classifier, trained on real and synthetic samples, to distinguish between the two classes. Lower classification accuracy suggests that the generated data is harder to differentiate from the real data, implying greater similarity between real and synthetic data. For this task, we trained logistic regression (LR), random forest (RF), and multi-layer perceptron (MLP) classifiers and evaluated their accuracy and F1 scores.

Utility measures how useful synthetic data are for downstream tasks. In our experiments, we trained a multi-class classifier on synthetic data to distinguish between data groups (e.g., tissues) and tested the performance of the model on real data. This validation procedure is commonly referred to as TSTR (Train on Synthetic, Test on Real). We considered three classifiers, namely LR, RF, and MLP. Since real datasets exhibit significant class imbalance, we evaluated performance using balanced accuracy and balanced F1 score metrics, which account for imbalance by weighting each class equally, ensuring that majority classes do not dominate the evaluation and providing more meaningful and comparable results.

#### 4.1.3. Statistical Tests

Statistical tests can quantitatively assess differences between individual features in a dataset, enabling a fine-grained analysis of specific characteristics. Specifically, given a dataset $X=\{x_{1},x_{2},\ldots ,x_{n}\}$ and a corresponding synthetic dataset $\hat{X}=\left\{{\hat{y}}_{1},{\hat{y}}_{2},\ldots ,{\hat{y}}_{m}\right\}$, we employ the Kolmogorov–Smirnov (KS) test to evaluate, via hypothesis testing, whether the feature distributions $P_{X}$ and $P_{\hat{X}}$ are significantly different.

#### 4.1.4. Visualization Techniques

Visualization techniques are projection-based metrics used for reducing higher-dimensional data into a lower-dimensional subspace. UMAP [25] is a non-linear dimension reduction technique. The method constructs a mathematical representation of the data topology through fuzzy simplicial sets, capturing both local and global structural relationships. UMAP’s optimization process minimizes the cross-entropy between high- and low-dimensional topological representations The resulting visualizations are presented in Appendix B.

### 4.2. Experiment Design

For both datasets, we implemented the same experimental procedure. First, we selected a set of covariates to condition the generation of the samples. Each covariate was treated as categorical, and the dimension of the embedding of each covariate was determined by the formula:$$d_{j}=\left\lfloor \sqrt{l_{j}}+1\right\rfloor$$
where $l_{j}$ is the vocabulary size of the *j*th covariate. We then compared the ability to generate synthetic data across three models, namely the conditional variational autoencoder (CVAE), GAN, and WGAN-GP, plus the proposed BioGAN, using both the variant with GraphConv and the one with H2GCN layers.

Regarding the CVAE, both the encoder and the decoder have a single hidden layer of 256 neurons and a latent space dimension of 64. The GAN and the WGAN-GP adopt a fully connected architecture for both the generator and the discriminator. Each network has two hidden layers of 256 neurons and uses the LeakyReLU activation function to cope with the vanishing gradient issue. For BioGAN, we adopted the same architecture of GAN and WGAN-GP for the discriminator, and we used four layers of message passing.

All the models were trained for 500 epochs using a batch size of 32 and the Adam optimizer with learning rates set to $\alpha =10^{-4}$, $\beta_{1}=0.5$, and $\beta_{2}=0.9$.

For *E. coli*, we conditioned the generation on several key experimental covariates. Following [13], we incorporated covariates that reflect critical environmental and experimental conditions during the cultivation of *E. coli*. Specifically, we included glucose concentration, ampicillin level, oxygen concentration, temperature, aeration, pH, and growth phase.

For *H. sapiens*, we used as covariates the tissue type, specific tissue subtype, patient age, and patient sex.

### 4.3. Results on E. coli Dataset

Table 1, reports the unsupervised performance metrics. It can be observed that the GAN generator achieves incredibly high values in *Precision* but performs very poorly in *recall* and *correlation*. This can be attributed to the well-known issue of mode collapse, a phenomenon in which the generator produces limited, repetitive outputs instead of diverse samples. Instead, the elevated values of *recall* and *correlation* of BioGAN, with both GraphConv and H2GCN layers, suggest that these models capture more complex patterns within the data, potentially due to their graph-based architectures, which may better align with the biological dependencies.

The results of the detectability tests are reported in Table 2. Most of the models score high F1 close to 1.0, indicating that the generated samples are easily distinguishable from real ones. However, BioGAN with the H2GCN layer stands out, achieving the lowest F1 score (0.841) for LR and MLP, indicating improved similarity to real data. In the principal component (PC) space, computed using the first 100 PCs, the detectability scores for GraphConv and H2GCN drop substantially. This improvement confirms that GraphConv and H2GCN produce synthetic data that are more similar to real data, making it harder for the classifier to distinguish between the two.

Table 3 reports the results of the TSTR (Train on Synthetic, Test on Real) utility test on the task of predicting the pH level at which the *E. coli* population was grown. The test set is highly imbalanced, with 85.8% of the samples belonging to a class with no corresponding pH value. Among the remaining samples, pH 7 is the most represented (7.1%), followed by pH 5.5 (2.2%), and pH 7.2 (2.2%). The other pH levels (2, 5.7, 8.7, 8.5, 5.3, and 7.6) each account for less than 1% of the data. A label with such imbalance and a percentage of missing values was intentionally selected to test the model’s ability to learn from unlabeled samples and to produce meaningful and exploitable samples for strongly underrepresented groups. Indeed, despite this imbalance, BioGAN with both GraphConv and H2GCN layers outperforms other models, particularly in LR and MLP, achieving scores close to 0.96 with a *p*-value computed with a *t*-test of $8.85e^{-22}$ and $3.10e^{-21}$, respectively. Such high performance suggests that the generated data retains key predictive features.

### 4.4. Results on GTEX Dataset

Here, we present the results on the GTEX dataset using tissue-specific coexpression matrices as prior knowledge. We also performed additional experiments integrating the co-expression networks with CollecTri network. The results of this analysis are available in Appendix C and Appendix D.

To further analyze the performance of the models, we also considered four additional datasets constructed by selecting only the 3000, 5000, and 10,000 genes with the highest distinctiveness, as well as the 3000 genes with the lowest distinctiveness. In this section, we report the comprehensive analyses on the 3000 most distinct and the full datasets, and the key performance indicators for the others.

#### 4.4.1. Unsupervised Metrics

The unsupervised metrics of the datasets comprising the 3000 most distinct genes, the 3000 less distinct genes, and all the genes are reported in Table 4. The baseline models (CVAE and GAN) demonstrated poor performance in all metrics and across the three datasets, indicating that they struggle to generate data that accurately reflects the real data distribution. The *precision* on the dataset of the 3000 least distinct genes is an exception; however, the associated low values of both *recall* and *correlation* indicate that the generated samples exhibit extremely low variability, most likely attributable to mode collapse. The WGAN-GP model introduced substantial enhancements, increasing all three metrics by more than 20%. BioGAN, with both GraphConv and H2GCN layers, demonstrated superior performance across all metrics and datasets. In particular, BioGAN with H2GCN and a correlation graph pruned at 0.9 achieved superior performance in five out of nine cases. In summary, BioGAN consistently outperformed the baseline, highlighting the advantages of incorporating the network structure into the generation process.

#### 4.4.2. Detectability

Table 5 presents the results of the detectability test. We evaluated the performance of different classifiers in distinguishing between real and generated data. We ran the test on the dataset consisting of the 3000 with the highest distinctiveness and on the full dataset, considering both all the features and the first 100 principal components. The results are reported in terms of F1 scores.

As expected, synthetic data generated via CVAE and GAN is easily detectable, even using a simple LR, in both full-feature and PC space. This indicates the poor quality of the generated samples. Similarly, WGAN-GP maintains high detectability (F1 > 0.98) in the complete feature space, although it shows a marked performance improvement in the PC space, particularly for LR. In contrast, BioGAN displays a significant enhancement in performance, as demonstrated by the significant drop in detectability. In particular, the GraphConv layer with a threshold of 0.90 showed the lowest detection value for the MLP (0.991) and also a decrease in detection in PCs for both RF and MLP. The H2GCN layer, particularly with a 0.90 threshold, achieves the most promising results, showing a substantial 21.2% decrease in detectability for LR in the full feature space while simultaneously delivering the best performance in the PC space.

#### 4.4.3. Utility

Table 6 presents the utility metrics for the TSTR evaluation, where the classifiers have been trained to predict the tissue type. The results confirm that graph-based approaches outperform traditional models. BioGAN, incorporating GraphConv layers on the dataset threshold at 0.80, achieves the highest performance across all classifiers, with F1 scores of 0.975, 0.957, and 0.967 for LR, RF, and MLP, respectively.

To test the significance of those improvements with respect to WGAN-GP, we ran a *t*-test obtaining the following *p*-value: $1.80\times 10^{-23}$ , $1.66\times 10^{-14}$ , and $1.01\times 10^{-10}$ , for LR, RF, and MLP, respectively. For what concerns the evaluation of all genes, we obtained superior performances for LR (*p* value = $1.56\times 10^{-12}$) and MLP (*p* value = $1.65\times 10^{-05}$).

#### 4.4.4. Results on Incremental Dataset

To evaluate the performance and scalability of BioGAN, we conducted a comparative analysis between WGAN-GP and our approaches using a co-expression with a 0.90 threshold. The evaluation spans four gene-set sizes: 3000, 5000, 10,000, and 19,075 genes, focusing on three unsupervised metrics and F1 detectability for LR. Figure 2 presents all relevant plots, showing the following measures:Precision (top left): Both variants of our model maintain consistently high precision as the gene count increases. While WGAN-GP’s control line shows a steady linear increase, its overall performance remains lower than our proposed approaches.Recall (top right): All models exhibit a declining trend with an increasing gene count. The GraphConv layer version shows a more gradual descent compared to the WGAN-GP and H2GCN layer versions that drops to approximately 0.3 at 19,075 genes. The H2GCN variant experiences a slightly steeper decline in recall, with a loss of approximately 10% more in performance compared to the WGAN-GP version.WD_2_ (bottom left): All three approaches show increasing Wasserstein scores with higher gene counts, following nearly parallel trajectories. Notably, our model with an H2GCN layer maintains slightly lower WS2 values throughout the range.Detectability (bottom right): The GraphConv variant shows increasing detectability as gene count rises, while WGAN-GP’s detectability decreases. The H2GCN layer version stabilizes around 0.80, demonstrating superior performance over WGAN-GP. This suggests our approach generates more realistic data even at higher dimensionalities.

## 5. Discussions and Conclusions

This study addresses the challenge of generating synthetic gene expression profiles for human subjects to support downstream analyses, aiming to overcome intrinsic genomic data issues such as scarcity, privacy concerns, heterogeneity across institutions, and demographic biases.

While existing data augmentation techniques fail to capture the complexity of biological data, we introduced the BioGAN model, a deep learning architecture that integrates the adversarial framework of WGAN-GP with GNNs to generate more realistic synthetic gene expression profiles. Our model was tested on two scenarios: the generation of expression profiles for *E. coli* using the gene regulatory network from RegulonDB, and generating human gene expression profiles from tissue-specific co-expression networks. The results demonstrate that incorporating a priori biological knowledge is an effective strategy to improve the quality and utility of synthetic transcriptomic data.

However, we also identified some limitations. As gene expression samples became more complex, performance declines across all models, including our architecture. This highlights a challenge in the field: maintaining the quality of synthetic data generation as dimensionality increases, which is particularly relevant for complex gene expression data.

### Future Directions

This work opens several promising research directions. One avenue involves enhancing biological network integration by leveraging heterogeneous graphs with multiple edge types and attributes, allowing for a more comprehensive representation of biological systems and enabling multi-omic data generation. This is especially relevant for GRNs, which are characterized by a high degree of sparsity in their connections, as previously highlighted. By integrating complementary omics data—such as transcriptomics, epigenomics, and proteomics—each represented with distinct node attributes, the generative model can capture a more comprehensive picture of cellular regulation. Furthermore, enriching the graph with diverse types of interactions—derived from sources like literature mining, large language models, or experimentally validated databases—can provide additional biological context and improve the quality of the generation. However, as graph complexity increases, so do computational demands, making scalability a critical challenge. Addressing this issue could involve exploring computationally efficient training strategies such as graph pruning, compression, and rewiring [26]. These techniques would help manage memory usage and computational costs while preserving essential biological relationships, ensuring that generated data remain both biologically meaningful and computationally feasible.

Another key direction is the development of biologically informed evaluation metrics that extend beyond conventional statistical measures. While current methods primarily assess the distributional similarity between real and synthetic data, future work could incorporate domain-specific constraints to evaluate the biological plausibility of generated genomic data. This would enhance the interpretability and reliability of synthetic data for downstream applications.

Additionally, systematic graph manipulations could provide deeper insights into gene–gene interactions and network robustness. For instance, computationally simulating gene knockout experiments by selectively removing nodes from the graph could help assess the impact of individual genes on expression patterns. Similarly, targeted or random edge modifications could reveal critical regulatory relationships and test the resilience of biological networks. Such analyses could also contribute to disease modeling by comparing network structures in healthy and pathological states, potentially identifying key genetic alterations and compensatory mechanisms associated with disease progression.

## Figures and Tables

**Figure 1 bioengineering-12-00658-f001:**
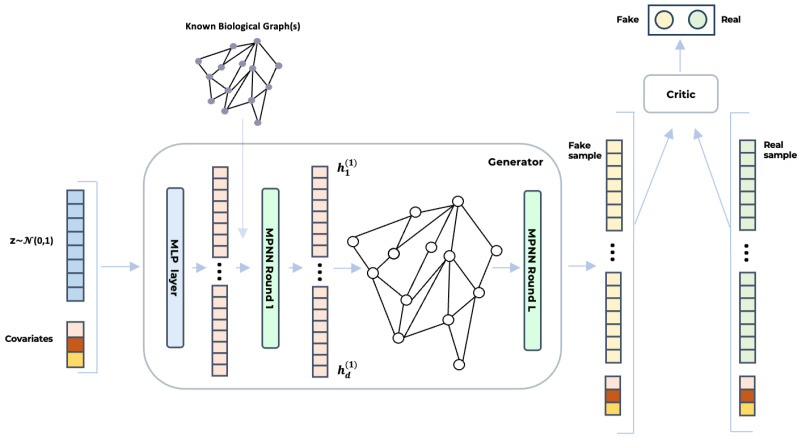
The figure illustrates the architecture of BioGAN for biologically informed gene expression synthesis. A noise vector, *z*, sampled from a normal distribution and concatenated with additional covariates, is first processed via an MLP. The output ($h^{\left(1\right)}$) is then passed through a series of MPNN layers (either GraphConv or H2GCN), which leverage known biological relationships—such as gene co-expression or regulatory networks—to generate synthetic gene expression profiles. Finally, the critic, implemented as an MLP, evaluates both real and synthetic samples ($h^{\left(L\right)}$) to distinguish between them, guiding the generator toward producing increasingly realistic data.

**Figure 2 bioengineering-12-00658-f002:**
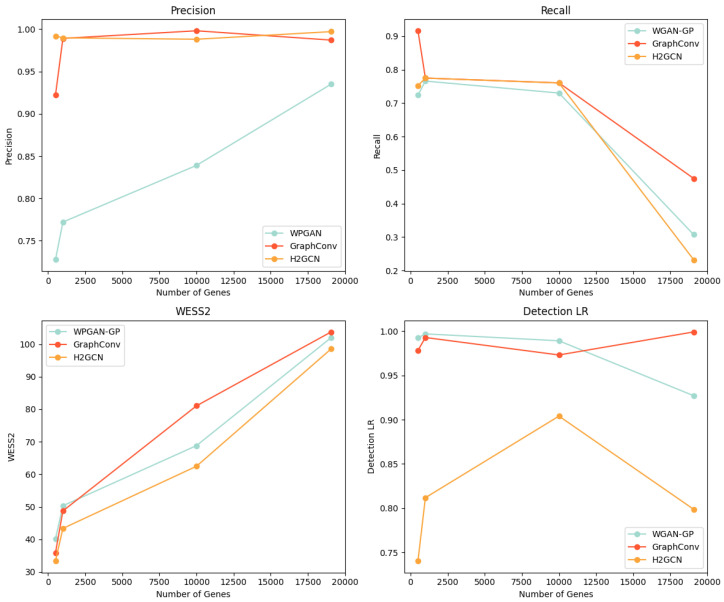
Comparison of WGAN-GP and the proposed models with GraphConv and H2GCN layers across three gene set sizes (3000, 5000, 10,000, and 19,075 genes), evaluating precision, recall, WS2, and F1 detectability for LR.

**Table 1 bioengineering-12-00658-t001:** Unsupervised performances for data generated by different models, with results averaged across 10 runs. The best scores are reported in bold.

Model	Precision (↑)	Recall (↑)	Correlation (↑)
CVAE	0.338 ± 0.038	0.041 ± 0.031	0.532 ± 0.005
GAN	**1.000 ± 0.000**	0.000 ± 0.000	0.019 ± 0.026
WGAN-GP	0.914 ± 0.003	0.620 ± 0.016	0.809 ± 0.011
BioGAN-GraphConv	0.515 ± 0.024	0.917 ± 0.008	**0.834 ± 0.006**
BioGAN-H2GCN	0.487 ± 0.024	**0.928 ± 0.010**	0.833 ± 0.008

**Table 2 bioengineering-12-00658-t002:** Detectability test results for the different models, evaluated considering both the full feature space and the first 100 PCs. The table reports the F1 scores achieved via logistic regression (LR), multi-layer perceptron (MLP), and random forest (RF) classifiers, averaged over 10 runs. Best performance in bold.

	Entire Features Space (↓)			Principal Components (↓)		
Model	F1 (LR)	F1 (RF)	F1 (MLP)	F1 (LR)	F1 (RF)	F1 (MLP)
CVAE	0.998 ± 0.001	1.000 ± 0.000	1.000 ± 0.000	0.988 ± 0.002	0.992 ± 0.003	1.000 ± 0.000
GAN	1.000 ± 0.0000	1.000 ± 0.0000	1.000 ± 0.0000	0.980 ± 0.003	0.999 ± 0.001	1.000 ± 0.0000
WGAN-GP	1.000 ± 0.0000	**0.994 ± 0.005**	1.0000 ± 0.000	0.847 ± 0.007	0.971 ± 0.004	0.995 ± 0.001
BioGAN-GraphConv	0.845 ± 0.026	1.000 ± 0.0000	0.940 ± 0.015	0.571 ± 0.024	**0.940 ± 0.012**	0.949 ± 0.007
BioGAN-H2GCN	**0.841 ± 0.013**	1.0000 ± 0.0000	**0.940 ± 0.010**	**0.569 ± 0.021**	0.942 ± 0.006	**0.942 ± 0.011**

**Table 3 bioengineering-12-00658-t003:** TSTR (Train on Synthetic, Test on Real) performance for data generated via different models in predicting the pH level. The table reports the F1 scores achieved through logistic regression (LR), multi-layer perceptron (MLP), and random forest (RF) classifiers, averaged over 10 runs. Best performance in bold.

Model	F1 (LR) (↑)	F1 (RF) (↑)	F1 (MLP) (↑)
CVAE	0.787 ± 0.059	0.647 ± 0.098	0.679 ± 0.036
GAN	0.417 ± 0.035	0.799 ± 0.000	0.296 ± 0.009
WGAN-GP	0.753 ± 0.010	**0.888 ± 0.002**	0.776 ± 0.006
BioGAN-GraphConv	0.962 ± 0.005	0.877 ± 0.009	**0.959 ± 0.0091**
BioGAN-H2GCN	**0.963 ± 0.006**	0.864 ± 0.008	0.958 ± 0.009

**Table 4 bioengineering-12-00658-t004:** Unsupervised metrics of different models on three datasets: one with the 3000 most distinct genes, one with less distinct genes, and one including all genes. Results are averaged across 10 runs. Best performance in bold.

Model	Precision (↑)	Recall (↑)	Correlation (↑)
3000 most distinct			
CVAE	0.221 ± 0.017	0.181 ± 0.037	0.429 ± 0.016
GAN	0.749 ± 0.009	0.233 ± 0.032	0.752 ± 0.002
WGAN-GP	0.727 ± 0.014	0.723 ± 0.010	0.936 ± 0.002
BioGAN-GraphConv (0.90)	0.922 ± 0.005	0.916 ± 0.006	0.973 ± 0.001
BioGAN-GraphConv (0.80)	0.669 ± 0.019	**0.926** ± 0.006	0.958 ± 0.001
BioGAN-H2GCN (0.90)	**0.991** ± 0.003	0.752 ± 0.015	0.981 ± 0.001
BioGAN-H2GCN (0.80)	0.988 ± 0.001	0.858 ± 0.009	**0.983** ± 0.001
3000 less distinct			
CVAE	0.379 ± 0.011	0.000 ± 0.000	0.320 ± 0.005
GAN	0.939 ± 0.003	0.000 ± 0.000	0.322 ± 0.001
WGAN-GP	0.931 ± 0.005	0.012 ± 0.003	0.632 ± 0.002
BioGAN-GraphConv (0.90)	0.928 ± 0.007	0.066 ± 0.008	0.705 ± 0.003
BioGAN-GraphConv (0.80)	0.866 ± 0.011	**0.124** ± 0.012	0.716 ± 0.003
BioGAN-H2GCN (0.90)	**0.998** ± 0.001	0.005 ± 0.000	**0.749** ± 0.001
BioGAN-H2GCN (0.80)	0.997 ± 0.001	0.002 ± 0.001	0.738 ± 0.001
All genes			
CVAE	0.761 ± 0.005	0.002 ± 0.001	0.606 ± 0.004
GAN	0.761 ± 0.006	0.002 ± 0.001	0.606 ± 0.004
WGAN-GP	0.935 ± 0.006	0.307 ± 0.009	0.938 ± 0.002
BioGAN-GraphConv (0.90)	0.987 ± 0.003	0.475 ± 0.013	0.944 ± 0.002
BioGAN-GraphConv (0.80)	0.942 ± 0.007	**0.621** ± 0.015	0.935 ± 0.001
BioGAN-H2GCN (0.90)	**0.997** ± 0.002	0.232 ± 0.013	**0.960** ± 0.001
BioGAN-H2GCN (0.80)	0.997 ± 0.001	0.002 ± 0.001	0.738 ± 0.001

**Table 5 bioengineering-12-00658-t005:** Detectability of different models on three datasets. The table reports the F1 scores achieved via logistic regression (LR), multi-layer perceptron (MLP), and random forest (RF) classifiers, averaged over 10 runs. Best performance in bold.

	Entire Features Space ↓			PC’s ↓		
Model	F1 (LR)	F1 (RF)	F1 (MLP)	F1 (LR)	F1 (RF)	F1 (MLP)
3000 most distinct						
CVAE	0.995 ± 0.001	1.000 ± 0.001	0.999 ± 0.001	0.921 ± 0.005	0.998 ± 0.001	0.998 ± 0.001
GAN	1.00 ± 0.00	0.997 ± 0.00	1.00 ± 0.00	0.939 ± 0.004	0.992 ± 0.003	0.998 ± 0.000
WGAN-GP	0.992 ± 0.001	0.988 ± 0.001	0.999 ± 0.001	0.556 ± 0.012	0.972 ± 0.003	0.996 ± 0.0017
BioGAN-GraphConv (0.90)	0.978 ± 0.003	0.999 ± 0.001	**0.991 ± 0.002**	0.567 ± 0.011	0.936 ± 0.006	0.991 ± 0.002
BioGAN-GraphConv (0.80)	0.999 ± 0.001	0.999 ± 0.001	0.999 ± 0.001	0.671 ± 0.007	**0.896 ± 0.006**	**0.985 ± 0.002**
BioGAN-H2GCN (0.90)	**0.766 ± 0.005**	**0.934 ± 0.006**	0.996 ± 0.001	**0.522 ± 0.009**	0.919 ± 0.006	0.995 ± 0.002
BioGAN-H2GCN (0.80)	0.875 ± 0.003	0.997 ± 0.001	0.994 ± 0.002	0.5359 ± 0.004	0.920 ± 0.009	0.994 ± 0.002
All genes						
CVAE	1.000 ± 0.000	1.000 ± 0.000	1.000 ± 0.000	0.990 ± 0.002	0.999 ± 0.001	0.998 ± 0.001
GAN	1.000 ± 0.000	1.000 ± 0.000	1.000 ± 0.000	0.991 ± 0.001	0.999 ± 0.001	0.998 ± 0.001
WGAN-GP	0.992 ± 0.001	**0.988 ± 0.001**	0.999 ± 0.001	0.541 ± 0.018	0.971 ± 0.002	0.989 ± 0.001
BioGAN-GraphConv (0.90)	0.999 ± 0.001	1.000 ± 0.000	**0.991 ± 0.002**	0.563 ± 0.006	0.950 ± 0.003	0.981 ± 0.002
BioGAN-GraphConv (0.80)	0.999 ± 0.001	1.000 ± 0.000	0.999 ± 0.0001	0.662 ± 0.008	0.959 ± 0.005	**0.978 ± 0.003**
BioGAN-H2GCN (0.90)	**0.798 ± 0.003**	1.000 ± 0.000	0.990 ± 0.003	**0.524 ± 0.010**	**0.922 ± 0.008**	0.987 ± 0.003

**Table 6 bioengineering-12-00658-t006:** TSTR performance of different models on two datasets. The table reports the F1 scores achieved through logistic regression (LR), multi-layer perceptron (MLP), and random forest (RF) classifiers, averaged over 10 runs. Best performance in bold.

Model	F1 (LR) (↑)	F1 (RF) (↑)	F1 (MLP) (↑)
3000 most distinct			
CVAE	0.921 ± 0.012	0.186 ± 0.008	0.705 ± 0.018
GAN	0.809 ± 0.001	0.842 ± 0.007	0.797 ± 0.010
WGAN-GP	0.925 ± 0.001	0.940 ± 0.003	0.799 ± 0.04
Proposed-GraphConv (0.90)	0.968 ± 0.002	0.939 ± 0.005	0.955 ± 0.0023
Proposed-GraphConv (0.80)	**0.975 ± 0.002**	**0.957 ± 0.004**	**0.967 ± 0.003**
Proposed-H2GCN (0.90)	0.958 ± 0.003	0.929 ± 0.004	0.9505 ± 0.002
Proposed-H2GCN (0.80)	0.966 ± 0.003	0.934 ± 0.003	0.960 ± 0.001
All genes			
CVAE	0.384 ± 0.02	0.435 ± 0.001	0.344 ± 0.016
GAN	0.389 ± 0.015	0.435 ± 0.001	0.345 ± 0.017
WGAN-GP	0.959 ± 0.003	**0.988 ± 0.003**	0.940 ± 0.011
BioGAN-GraphConv (0.90)	0.975 ± 0.003	0.917 ± 0.003	0.951 ± 0.006
BioGAN-GraphConv (0.80)	**0.976 ± 0.001**	0.928 ± 0.009	**0.965 ± 0.008**
BioGAN-H2GCN (0.90)	0.967 ± 0.003	0.874 ± 0.006	0.949 ± 0.006

## Data Availability

Data is contained within the article.

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
