# Peer review of "BioGAN: Enhancing Transcriptomic Data Generation with Biological Knowledge"

_bioengineering, 2025, doi:10.3390/bioengineering12060658_

Round 1

Reviewer 1 Report

Comments and Suggestions for Authors

The manuscript presents some technique for transcriptome sequencing data modeling. It is computer simulation of so called ‘synthetic data’.
This work is superficial one lacking real application to biological problems. I’m afraid it does not fit the journal scope.
The figure 2 is not relevant (just data description not related to the analysis).
Figure 1 – it shows main scheme of research – should be updated.
Later figures (that looks like single cell analysis data) should be commented – what one should see there – clustering, tight group of dots, what? Why is such a presentation better than another?
Add detailed comments in the figure legend.
As a biologist I don’t see any sense in that completely artificial visualization.
The text has multiple formulas without proper legend (parameters explanation). And it is not used in the next text. 
The methods to compare in the tables are not described.
Two methods are from this work – BioGAN- H2GCN and BioGAN -GraphConv.
In the title only one method BioGAN claimed. And the results are better in one case or in another case. 
There are applications only for transcriptomics data set, but the claim is about all the genomics data simulation. 
The references are not complete.

The Abstract should clearly state the problem. These wording is too common, and unclear “Synthetic data generation offers a promising solution to these issue. However, existing approaches often fail to incorporate biological knowledge, limiting the realism and utility..” – what is the solution? What is the problem? What are these ‘existing approaches’? Need to specify it especially for biologists. 
Keywords list is too short. Add at least two more keywords to be specific.

By lines:
Line 17: “Artificial Intelligence (AI)-based solutions … have proven..” – need a reference to this prove. It is a claim. Who proved it?
Line 29: “GDPR .. and HIPAA..” – give the abbreviations in full.
Line 39: “Wasserstein Generative Adversarial Network” – give reference to it. Who is Wasserstein, why a such title.

The Background section should start from biological examples.

Line 69: - formula could be from new line, explain parameters k, and n.
Line 71: “Several popular generative models…” – give references to these models. Why it is ‘popular’?
Line 83: ‘first term represents’ – the formalism is not complete, parameters are not shown in the text. If these parameters were not used in the next part of the text, not need give a formula.
The same remarks are true for next formulas. Might be it is good as mathematical work, but not clear here.
The Related Works section.
Give the references from the beginning.
Give abbreviations GOGGLE in full.
Show an example of application.
Figure 1 – give MPNN abbreviation in full.
Line 259: ‘M 3D’ – how this formula appeared – is it in power 3D, or 3D is three-dimensional?
Line 262: ‘M3D database’ – cite this database properly, if this is a database.
Line 263: “7,459 probes .We considered only the 2,648 probes” -  why these numbers, what it means? I understand that it is microarray, and probes should be related to genes. If used less probes – need comment.
There is no reference.
Line 264: ‘RegulonDB database’ – need cite this database, give reference.
Line 273: ‘GTEx’ need reference to this platform, comment what is it.
Line 281: “We used two thresholds, 0.8 and 0.9, respectively. We also considere a Gene Regulatory Network, derived from the union of the DoRothEA and CollecTri  datasets.” – here is not clear what is the threshold, for what?

‘consider’ – typo.
How the network was constructed? There are many methods to reconstruct a network.
Comment on DoRothEA and CollecTri databases, give abbreviation in full. What kind of data are there?
Section 4 – should be titled ‘Computational experiments’
Subsection ‘Precision and Recall’ not necessary here, standard formulas, not need to describe. Just give a reference to statistics textbook or similar publication for these metrics.
Standard deviation, correlation – all these are too common terms to describe in main text.
Line 428: ‘4.4.3. Utility’ – this section is unclear – utility for what? How it should be used. 
What is ‘TSTR evaluation’?
Line 440: “It is clear that in both the CVAE and GAN models..” – it is absolutely not clear!
What one can see in Figure 3? The same dots, with colors in the same shape.
(by the way, I’m not color blinded, I see different colors, green, orange – on left panel, and only blue and yellow on right panel. But it is hard to see for other people, some readers are color-blinded).
The shape of so called fake points resemble the shape of cluster of ‘blue’ real points. So, where is the difference, how to see that one method is better than another? In producing fake data?
Sorry, maybe it really shows some real correlation in gene expression data, but this figure is not understandable for a reader.
The discussion section has common phrases. 
I believe it is not necessary to make more complex common network methods, but it is better develop applications for different models – for yest, for human gene expression to capture gene structure, develop some computer simulation technique that gives practical results (in statistical estimates etc.)
The references should be formatted properly – see 4,5,15 – mix author names and abbreviation (consortia names)
“GDPR, G.D.P.R. G”
“Act, A.”
“Consortium, G.”

Author Response

R: The manuscript presents some technique for transcriptome sequencing data modeling. It is computer simulation of so called ‘synthetic data’.

This work is superficial one lacking real application to biological problems. I’m afraid it does not fit the journal scope.

A: We thank the reviewer for their direct and detailed feedback. However, we respectfully disagree with the critique. The generation of synthetic data plays a pivotal role in enabling the development and validation of Artificial Intelligence methods across a wide range of domains, including but not limited to biology and healthcare.  In biomedical research, the importance of high-quality synthetic data is even more pronounced due to strict privacy regulations (such as GDPR in Europe), which often limit the availability and sharing of real-world patient data. Synthetic data offers a viable and ethically sound alternative, enabling data-driven innovation while preserving patient confidentiality. Furthermore, synthetic data generation can facilitate data augmentation, helping to mitigate biases arising from underrepresented patient populations.  Our experiments provide empirical evidence that the data generated by our framework can be effectively used to train AI models for downstream tasks. These results underscore the practical utility and relevance of our approach.  Lastly, we would like to highlight that this manuscript was submitted to the special issue “Computational Genomics for Disease Prediction,” to which it is highly relevant, both in scope and in scientific contribution.

R: The figure 2 is not relevant (just data description not related to the analysis).

A: We agree with the reviewer that Figure 2 emphasizes a secondary aspect of the dataset that is not directly relevant to the main analysis. In response, we have moved this figure to the Appendix to maintain the focus of the main text.

R: Figure 1 – it shows main scheme of research – should be updated.

A: We thank the reviewer for their comment regarding Figure 1. However, we are not entirely sure about the specific nature of the requested update. We would like to clarify that Figure 1 is not intended to represent the entire research workflow, but rather to illustrate the artificial intelligence method that we are presenting and testing. We believe that the content of the figure is both appropriate and coherent with the methodology presented in the manuscript. Furthermore, the format and “graphical syntax” used in the figure follow standard conventions commonly adopted for this type of visualization in applied computational sciences.

R: Later figures (that looks like single cell analysis data) should be commented – what one should see there – clustering, tight group of dots, what? Why is such a presentation better than another?

Add detailed comments in the figure legend.

As a biologist I don’t see any sense in that completely artificial visualization.

A: We thank the reviewer for raising the important point regarding the clarity and interpretability of the figures. As these visualizations may not be easily interpretable to a broader audience, particularly those without a computational or machine learning background, we have opted to move them directly to the supplementary material (Figure A2). This decision was made to avoid confusion in the main text while still making the visualizations available for interested readers.

R: The text has multiple formulas without proper legend (parameters explanation). And it is not used in the next text. 

The methods to compare in the tables are not described.

A: According to reviewer suggestion we revised the text of the method section to make sure that all the parameters and symbols in the formulas have been properly introduced.
All the methods presented in the tables have been either extensively described (BioGAN in its two variants) or properly cited (VAE, GAN, and WGAN-GP).

R: There are applications only for transcriptomics data set, but the claim is about all the genomics data simulation. 

The references are not complete.

A: The title and the abstract of the paper clearly state that our method is about the generation (NOT simulation) of transcriptomic data. We mentioned genomic data in the introduction, to motivate our research in a broader context. Therefore, we believe that there is not any overclaiming.

R: The Abstract should clearly state the problem. These wording is too common, and unclear “Synthetic data generation offers a promising solution to these issue. However, existing approaches often fail to incorporate biological knowledge, limiting the realism and utility..” – what is the solution? What is the problem? What are these ‘existing approaches’? Need to specify it especially for biologists. 

Keywords list is too short. Add at least two more keywords to be specific.

We thank the reviewer for their thoughtful comments regarding the abstract and keywords. With respect to the abstract, we respectfully note that the identified issues are already addressed in the current version. Specifically, the sentence “Yet, challenges such as data scarcity, privacy constraints, and biases persist” clearly outlines the problem, while the subsequent sentence—“In this study, we introduce BioGAN, a novel biologically informed generative model that integrates Graph Neural Networks within a Conditional Wasserstein GAN with Gradient Penalty…”—clearly introduces our proposed solution. In response to the reviewer’s suggestion, we have clarified in the abstract that the “existing approaches” refer to methods based on generative artificial intelligence. However, we have chosen not to elaborate further on these methods within the abstract, as they are not the central contribution of the manuscript; more details are provided in Section 2 . Finally, in accordance with the reviewer’s recommendation, we have expanded the keywords list to include two additional, more specific terms. The current list follows:

Synthetic Transcriptomic Data; Generative Adversarial Networks; Graph Neural Networks; Generative Artificial Intelligence; Biologically Informed Methods; Computational Biology.

R: By lines:

R: Line 17: “Artificial Intelligence (AI)-based solutions … have proven..” – need a reference to this prove. It is a claim. Who proved it?  

A: Although we believe that, in the context in which it appears, the mentioned sentence is not a claim but rather an observation of current research trends—supported by thousands of publications—we have revised the sentence to soften its tone.

R: Line 29: “GDPR .. and HIPAA..” – give the abbreviations in full.

A: Modified.

R: Line 39: “Wasserstein Generative Adversarial Network” – give reference to it.

A: We added both the reference for Wassertein GAN and for the improved version  with gradient penalty.

R: Who is Wasserstein, why a such title.

A: This is already explained in section 2.1.2. “The Wasserstein GAN (WGAN) addresses this issue by replacing the standard loss with the Wasserstein distance, leading to more stable gradients.”

R: The Background section should start from biological examples.

A: We thank the reviewer for their suggestion. However, we believe that structuring the Background section around the key concepts—rather than a specific biological examples—is more appropriate for introducing and contextualizing our contribution. While we understand that preferences may vary, we are not aware of any publication standard or journal requirement that mandates starting with biological examples. In our view, including such examples at that stage would not serve the purpose of clarifying the methodological framework that underpins our work.

R: Line 69: - formula could be from new line, explain parameters k, and n.

A: We thank the reviewer for highlighting this issue. We added an explaination of the two parameters.

R: Line 71: “Several popular generative models…” – give references to these models. Why it is ‘popular’?

A: According to the suggestion of the reviewer,  we revised the sentence as:

“Deep learning generative models, such as Variational Autoencoders (VAEs) [Kingma, 2019] and Generative Adversarial Networks (GANs) [Mirza, 2014], perform well at this task.”

removing the inappropriate “popular” adjective and adding the two references.

R: Line 83: ‘first term represents’ – the formalism is not complete, parameters are not shown in the text. If these parameters were not used in the next part of the text, not need give a formula.

The same remarks are true for next formulas. Might be it is good as mathematical work, but not clear here.

We thank the reviewer for their observations. We would like to clarify that  “first term” and “second term” refer directly to the two additive components in the immediately preceding formula. Regarding the comment, “Might be it is good as mathematical work, but not clear here,” we are unsure of the specific concern being raised.

R: The Related Works section. Give the references from the beginning.

A: Modified in the text.

R: Give abbreviations GOGGLE in full.

A: Modified in the text.

R: Show an example of application.

Figure 1 – give MPNN abbreviation in full.

R: Line 259: ‘M 3D’ – how this formula appeared – is it in power 3D, or 3D is three-dimensional? 

Line 262: ‘M3D database’ – cite this database properly, if this is a database.

A: Modified in the text.

R: Line 263: “7,459 probes .We considered only the 2,648 probes” -  why these numbers, what it means? I understand that it is microarray, and probes should be related to genes. If used less probes – need comment.

A: In the current version of the manuscript it is already specified how those probes have been selected: “We considered only the 2,648 probes that are part of at least one regulatory interaction in the RegulonDB database”

There is no reference.

R: Line 264: ‘RegulonDB database’ – need cite this database, give reference.

A: Modificied in the text.

R: Line 273: ‘GTEx’ need reference to this platform, comment what is it.

A: Modified in the text.

R: Line 281: “We used two thresholds, 0.8 and 0.9, respectively. We also considere a Gene Regulatory Network, derived from the union of the DoRothEA and CollecTri  datasets.” – here is not clear what is the threshold, for what?

‘consider’ – typo. 

A: The intent of the threshold is specified in the sentence immediately preceding the reported one: “The correlation matrix has been filtered to retain only those values whose absolute value is above a predetermined threshold.”

R: How the network was constructed? There are many methods to reconstruct a network.

R: Comment on DoRothEA and CollecTri databases, give abbreviation in full. What kind of data are there?

A: We thank the reviewer for their observation and the opportunity to clarify. As stated in the manuscript, the E. coli regulatory network was retrieved from RegulonDB, a curated database of transcriptional regulation in E. coli. For the human dataset, the network was constructed based on the correlation of gene expression values. The reviewer is correct in noting the mention of DoRothEA and CollecTRI—these were included by mistake. Only CollecTRI was in fact used in a separate experiment presented in the Appendix, but it is not part of the main analysis.

R: Section 4 – should be titled ‘Computational experiments’

A: Modified in the manuscript.

R: Subsection ‘Precision and Recall’ not necessary here, standard formulas, not need to describe. Just give a reference to statistics textbook or similar publication for these metrics.

A: We thank the reviewer for this remark. We understand the possible confusion, as the metrics referred to in this subsection share the same names—Precision and Recall—as standard supervised learning metrics, although they are conceptually different. These metrics were not introduced by us, but are commonly used in the context of evaluating generative models. Given their substantial differences from the standard definitions, we believe that including their explicit description is important to avoid ambiguity and ensure clarity for all readers.

R: Standard deviation, correlation – all these are too common terms to describe in main text.

A: While we agree that standard deviation and correlation are commonly used statistical terms, in our work they are applied in a specific and non-trivial context. For this reason, we believe it is important to briefly explain their usage in the main text to avoid ambiguity and to ensure clarity for readers who may not be familiar with their application in this particular setting.

R: Line 428: ‘4.4.3. Utility’ – this section is unclear – utility for what? How it should be used. 

We thank the reviewer for pointing this out. The utility metric is introduced in the preceding section. In response to the reviewer’s suggestion, we have clarified this connection by updating the title of that subsection from “Supervised Metrics”to “Supervised Metrics – Detectability and Utility”, making it easier for the reader to locate and understand the definition and purpose of the metric.

R: What is ‘TSTR evaluation’?

A: We thank the reviewer for identifying this point. While the evaluation procedure is explained in the appropriate section of the manuscript, we acknowledge that it was not explicitly referred to using the commonly adopted name. In response, we have updated the text to include the sentence: “This validation procedure is commonly referred to as TSTR (Train on Synthetic, Test on Real).” to improve clarity and consistency with established terminology.

R: Line 440: “It is clear that in both the CVAE and GAN models..” – it is absolutely not clear!

What one can see in Figure 3? The same dots, with colors in the same shape.

(by the way, I’m not color blinded, I see different colors, green, orange – on left panel, and only blue and yellow on right panel. But it is hard to see for other people, some readers are color-blinded).

The shape of so called fake points resemble the shape of cluster of ‘blue’ real points. So, where is the difference, how to see that one method is better than another? In producing fake data?

Sorry, maybe it really shows some real correlation in gene expression data, but this figure is not understandable for a reader.

A: We thank the reviewer for their detailed feedback. We agree that the interpretation of Figure 3 may not be straightforward. Dimensionality reduction techniques such as UMAP, which we used in this figure, are commonly employed to provide an intuitive visual assessment of whether a generative model captures the structure of the original data distribution. However, we acknowledge that such visualizations can be difficult to interpret unambiguously and may not effectively communicate differences between models, especially for readers unfamiliar with these techniques or with color vision deficiencies. Following the reviewer’s concerns, and to avoid potential confusion, we have decided to remove this visualization-based validation from the main manuscript and include it instead in the appendix, where it can still be consulted for completeness.

R: The discussion section has common phrases. 

I believe it is not necessary to make more complex common network methods, but it is better develop applications for different models – for yest, for human gene expression to capture gene structure, develop some computer simulation technique that gives practical results (in statistical estimates etc.)

A: We thank the reviewer for their thoughtful comment. While we appreciate the suggestion to focus on developing applications or simulation techniques with direct practical results, we respectfully note that this reflects a different research objective from the one we pursue. Our aim is to explore and advance the foundational aspects of generative modeling methods for transcriptomic data, which we believe is equally important to enable downstream applications—including those suggested by the reviewer. We thus consider our contribution complementary rather than alternative, and we do not believe the suggested direction necessarily constitutes a stronger or more appropriate claim for our specific research goals.

R:The references should be formatted properly – see 4,5,15 – mix author names and abbreviation (consortia names)

“GDPR, G.D.P.R. G”

“Act, A.”

“Consortium, G

 A: Modified in the text

Reviewer 2 Report

Comments and Suggestions for Authors

Comments

While the integration of GNNs into GANs for synthetic omics generation is a promising concept, the manuscript doesn't convincingly explain how BioGAN provides substantial biological interpretability or novelty over prior models such as GOGGLE or attention-based GANs. Clarify how the proposed use of GraphConv and H2GCN architectures leads to deeper biological relevance beyond preserving statistical properties. Include a comparative interpretability discussion.

The manuscript heavily focuses on mathematical/statistical similarity between real and generated data but lacks any experimental or literature-backed biological validation (e.g., pathway enrichment, biological coherence of synthetic samples). Include biological relevance assessments such as pathway/pathway co-expression preservation or top DEGs between real and synthetic datasets.

The extensive metric-based evaluation (Precision, Recall, F1, WD2) is technically robust but lacks a discussion on the practical biological interpretability of the synthetic data. Add examples illustrating the biological meaning of low detectability or high utility scores, perhaps by demonstrating downstream use in differential expression or gene clustering.

The authors report results averaged over 10 runs but do not perform any statistical testing (e.g., t-test, ANOVA) to verify if improvements are statistically significant. Include p-values and confidence intervals when comparing BioGAN with baseline methods to validate the claims more rigorously.

The manuscript uses both GRNs and GCNs but doesn’t systematically compare the effect of using each graph type on performance. Provide a side-by-side comparative analysis showing how GRN-informed generation differs in structure and realism from GCN-informed generation.

The manuscript mentions using 0.8 and 0.9 correlation thresholds for co-expression networks but lacks justification or biological grounding for these cutoffs. Discuss why these thresholds were chosen, their biological relevance, and test sensitivity of results to different thresholds.

Given the use of graph-based models and large-scale omics data, computational complexity and resource usage are critical, but the manuscript is silent on this. Report training time, memory requirements, and computational scalability of BioGAN versus other models.

The implementation details of the GraphConv and H2GCN layers are vaguely explained, particularly the update rules and handling of heterophily. Include clear pseudocode or visual schematics for how node features are updated across layers in both GraphConv and H2GCN, especially in the context of biological graphs.

There is no mention of code availability, and several key hyperparameters (dropout rate, weight initialization, etc.) are missing. Make the model code publicly available, and provide a table of hyperparameters with detailed architectural information to facilitate reproducibility.

Although the authors mention label imbalance in E. coli pH values, they do not discuss label imbalance in GTEx tissue classes, which may significantly affect classification utility scores. Include a class distribution analysis for GTEx tissues and possibly re-evaluate using stratified sampling or data augmentation techniques.

Author Response

R: While the integration of GNNs into GANs for synthetic omics generation is a promising concept, the manuscript doesn't convincingly explain how BioGAN provides substantial biological interpretability or novelty over prior models such as GOGGLE or attention-based GANs. Clarify how the proposed use of GraphConv and H2GCN architectures leads to deeper biological relevance beyond preserving statistical properties. Include a comparative interpretability discussion.

The manuscript heavily focuses on mathematical/statistical similarity between real and generated data but lacks any experimental or literature-backed biological validation (e.g., pathway enrichment, biological coherence of synthetic samples). Include biological relevance assessments such as pathway/pathway co-expression preservation or top DEGs between real and synthetic datasets.

The extensive metric-based evaluation (Precision, Recall, F1, WD2) is technically robust but lacks a discussion on the practical biological interpretability of the synthetic data. Add examples illustrating the biological meaning of low detectability or high utility scores, perhaps by demonstrating downstream use in differential expression or gene clustering.

A: We thank the reviewer for this important comment. We would like to clarify that our primary objective is not to produce biologically interpretable synthetic data, but rather to generate data that closely resemble real omics data and are useful for downstream tasks. The incorporation of prior biological knowledge in our model serves a utilitarian purpose—to improve the realism and utility of the generated data. In contrast to GOGGLE, our approach requires significantly fewer parameters, making it scalable to high-dimensional datasets, whereas GOGGLE is limited to a few hundred features due to its architectural complexity. Additionally, prior work by Lacan et al. has shown that attention-based GANs often yield suboptimal performance in this context. Unlike our graph-based approach, attention mechanisms learn interactions without constraints and do not incorporate structured biological priors, which we believe is a key limitation.

R:The authors report results averaged over 10 runs but do not perform any statistical testing (e.g., t-test, ANOVA) to verify if improvements are statistically significant. Include p-values and confidence intervals when comparing BioGAN with baseline methods to validate the claims more rigorously.

 A: We thank the reviewer for the helpful suggestion. In response, we have incorporated statistical testing into our evaluation by reporting p-values, focusing specifically on the utility-related metrics. We believe these are the most relevant for our application, as they directly reflect the downstream performance and practical impact of the generated data.

R: The manuscript uses both GRNs and GCNs but doesn’t systematically compare the effect of using each graph type on performance. Provide a side-by-side comparative analysis showing how GRN-informed generation differs in structure and realism from GCN-informed generation.

A: We appreciate the reviewer’s insightful comment regarding the comparative role of gene regulatory GRNs and GCNs in our model. While we initially explored the use of GRNs alone, we found that the available regulatory network was too sparse to sufficiently capture the range of gene–gene interactions required for effective representation learning. In particular, the limited edge density in the GRN resulted in insufficient information propagation during message passing, leading to suboptimal model performance.

For this reason, we chose to fuse the GRN with a more densely connected co-expression graph, leveraging the complementary strengths of both sources. Our analysis therefore focuses on evaluating the impact of this graph fusion strategy, rather than a direct comparison between the individual graph types.

We agree that a side-by-side comparison would be valuable in other contexts, but given the sparsity and limited standalone performance of the GRN in our setting, we opted not to include it as an isolated baseline. Future work could explore the integration of more comprehensive or experimentally derived GRNs to assess their standalone utility more robustly.

R: The manuscript mentions using 0.8 and 0.9 correlation thresholds for co-expression networks but lacks justification or biological grounding for these cutoffs. Discuss why these thresholds were chosen, their biological relevance, and test sensitivity of results to different thresholds.

A: We thank the reviewer for this important observation. The choice of high correlation thresholds (0.8 and 0.9) for constructing co-expression networks was motivated by both biological and computational considerations. From a biological standpoint, these thresholds ensure that only the strongest and most meaningful gene–gene associations are retained, allowing each node to be influenced primarily by its most relevant neighbors. This aligns with the intuition that, in gene co-expression analysis, high correlation values are more likely to reflect functionally or regulatory meaningful relationships.

From a computational perspective, using sparser graphs derived from higher thresholds results in a significantly reduced number of edges, which in turn leads to lighter models in terms of parameter count and training time. This sparsity allows the model to focus on the most informative connections, improving interpretability and potentially enhancing generalization by avoiding overfitting to noisy or weak interactions.

We also added the following clarification to the text in Section 3.2.2.

We selected two correlation thresholds, 0.8 and 0.9, based on both biological and computational considerations. Biologically, these high thresholds ensure that only the strongest and most meaningful gene–gene associations are retained. By doing so, each gene is influenced primarily by its most relevant neighbors, aligning with the understanding that high correlation values are more likely to represent functionally or regulatory significant relationships in gene co-expression analysis.

From a computational perspective, applying higher thresholds results in sparser graphs with fewer edges, leading to models that are more efficient in terms of parameter count and training time. This sparsity enables the model to focus on the most informative connections, improving both interpretability and generalization. It also helps to mitigate the risk of overfitting to weak or noisy interactions, ultimately contributing to more robust predictions.

R: Given the use of graph-based models and large-scale omics data, computational complexity and resource usage are critical, but the manuscript is silent on this. Report training time, memory requirements, and computational scalability of BioGAN versus other models.

 A: We appreciate your valuable comment regarding the computational complexity and resource usage of our models. Unfortunately, we did not save the specific memory and training time data for the experiments. Additionally, the tests were conducted on different environments and GPUs, which introduces some variability in the results. However, based on our observations, we can report that BioGAN with GNN requires approximately 3 to 5 times more computational resources (in terms of training time) compared to WGAN-GP. This is primarily due to the additional complexity introduced by the graph-based architecture of BioGAN, which involves processing larger and more intricate relational data. We recognize the importance of these considerations and plan to investigate them further in future work, ensuring more precise tracking of computational requirements across different models.

R: The implementation details of the GraphConv and H2GCN layers are vaguely explained, particularly the update rules and handling of heterophily. Include clear pseudocode or visual schematics for how node features are updated across layers in both GraphConv and H2GCN, especially in the context of biological graphs.

 A: We thank the reviewer for the insightful comment. We appreciate the suggestion to provide more detailed explanations of the GraphConv and H2GCN layers.

In our implementation, we opted to leverage Geometric PyTorch (PyG), which is a highly optimized library for GNNs. Therefore, rather than implementing the GNN layers from scratch, we used the pre-built functions provided by PyG, which are widely used.

The semantics of the node feature updates, for both GraphConv and H2GCN, are indeed captured by the formulas for the node embedding update rule, which are reported in Section 3.1.1 of the manuscript.

In response to your suggestion, we have added a reference to the Geometric PyTorch implementation in the revised text to provide clarity on the underlying implementation of the GNN layers.

We added this paragraph to the manuscript:

All the deep learning models that we benchmarked were implemented using Python (version 3.10), PyTorch (version 2.3.1), and PyTorch Geometric (version 2.5.3) for GNNs. Specifically, for the graph convolutional layers we used GraphConv and the orginal code of H2GCN.

R: Although the authors mention label imbalance in E. coli pH values, they do not discuss label imbalance in GTEx tissue classes, which may significantly affect classification utility scores. Include a class distribution analysis for GTEx tissues and possibly re-evaluate using stratified sampling or data augmentation techniques.

 A: We thank the reviewer for pointing out the importance of class imbalance in the GTEx tissue classification task. We had originally included a figure illustrating the dataset distribution, which was later moved to the appendix (Figure A1). While human data indeed presents class imbalance, we addressed this issue by evaluating performance using balanced accuracy and balanced F1-score, which are more robust to such imbalances.  We agree with the reviewer that this was not sufficiently clear in the manuscript, and we have now clarified this by adding the following sentence to Section 4, Computational Experiments:

Since real datasets exhibit significant class imbalance, we evaluated performance using balanced accuracy and balanced F1 score metrics, that account for imbalance by weighting each class equally, ensuring that majority classes do not dominate the evaluation and providing more meaningful and comparable results.

Reviewer 3 Report

Comments and Suggestions for Authors

To summarize the paper - authors present a system which can generate artificial data which is very similar to real datasets and cannot be distinguished from real by their system. Clearly their system somehow averages and uses real data sets to generate new ones,  and obviously such data may be very similar to the real. Unclear why the data cannot be slightly modified ( add some noise) which will deliver the same results  - undetectable from real data, but will be much simpler. In any case all such kind of manipulations including AI, is only a kind of averaging and mixing. Which in no way will add in information content. 

Overall the use of such data is unclear. And of course it cannot substitute the generation of real data.  It would contradict today's concept of Individualized medicine etc. 

This is of high concern: “The test set is highly imbalanced, with 85.8% of the samples belonging to a class with no corresponding pH value. Among the remaining samples, pH 7 is the most represented (7.1%), followed by pH 5.5 (2.2%), and pH 7.2 (2.2%).”  - The authors used dataset where 85% has no assignment to pH to predict pH, sounds very suspicious. 

Motivation is another critical point, 

Synthetic data generation is usually used to fill in missing data in a set, mostly by inserting a most common value or averaged value. Here authors suggest to generate completely new data sets based on a bunch of other data sets.  The biological value of such data sets is unclear.  The reasoning that these data sets cannot be distinguished from the real sets would not give any biological meaning to such data. 

Author Response

Please find the response in the enclosed PDF file.

Round 2

Reviewer 1 Report

Comments and Suggestions for Authors

Thanks for the manuscript update. All the major comments were considered.

Though I believe that the synthetic data generation is not a real problem worthy for publication at biological journal, I have no more critiques to the authors regarding the text.

Minor comments:

Lines 87-88: in the formula  - parameter E should be commented (as reconstruction error) or entropy.

Line 88 – text after the formula (from word ‘where’ should start from left border), no indent space.

Line 95 and below:

Please comment on sign ~ - is it ‘aiming to’, or ‘approximately equal’?

Line 100: ‘Wasserstein GAN’ – add reference to this term (assume ref.[6]).

Line 103: start text after the formula without space left (Align to the left)

Line 109 and 138 – align text after the formula to the left. See also line 332.

The ‘WGAN-GP’ abbreviation first was given in line 108. Give it in full, and may repeat it in line 157 and 201. New term used throughout the text should be shown in full.

Section ‘2.2.1. Biological Graphs’ needs at least a reference. It has common definition of expression network, but it is rather oversimplified. Add a reference to this section to the definitions.

Lines 366 and 385 – section titles should not be in Italic font.

Line 368: CVAE abbreviation should be given in full (only VAE abbreviation was shown).

Line 381 – why ‘glucose concentration’ is in bold font, and next parameters are in Italic font?

The Tables tittles should be written without abbreviations.

Table 2 – rite ‘PCs’ in full.

Table 3 – write ‘TSTR’ in full

The references 4,5, 19 are still not formatted (The titles are mixed with the authors names initials)

Author Response

R: Thanks for the manuscript update. All the major comments were considered. 

Though I believe that the synthetic data generation is not a real problem worthy for publication at biological journal, I have no more critiques to the authors regarding the text. 

A: We sincerely thank the reviewer for their time, thoughtful feedback, and constructive suggestions. We appreciate that all major comments were considered. 

Minor comments: 

R: Lines 87-88: in the formula  - parameter E should be commented (as reconstruction error) or entropy. 

A: Modified in the text 

where the first term represents the reconstruction error (with E denoting the expected value under the approximate posterior q(z|x) 

R: Line 88 – text after the formula (from word ‘where’ should start from left border), no indent space. 

A: Modified in the text 

R: Line 95 and below:  

    Please comment on sign ~ - is it ‘aiming to’, or ‘approximately equal’? 

A: Modified in the text 

where ~  denotes sampling from a distribution 

R: Line 100: ‘Wasserstein GAN’ – add reference to this term (assume ref.[6]). 

A: Modified in the text 

R: Line 103: start text after the formula without space left (Align to the left) 

 Line 109 and 138 – align text after the formula to the left. See also line 332. 

A: Modified in the text  

R: The ‘WGAN-GP’ abbrviation first was given in line 108. Give it in full, and may repeat it in line 157 and 201. New term used throughout the text should be shown in full. 

A: Modified in the text  

R: Section ‘2.2.1. Biological Graphs’ needs at least a reference. It has common definition of expression network, but it is rather oversimplified. Add a reference to this section to the definitions. 

A: We thank the reviewer for the comment. We have revised the definitions in Section 2.2.1 to provide greater clarity and depth, and we have also added appropriate references to support them. 

Given the complexity of biological systems, and in particular the relationships among their components, graphs offer a convenient means of representation. Gene Co-expression Networks (GCN) and Gene Regulatory Networks (GRN) have been widely adopted to capture different types of interactions derived from transcriptomic data [10,11]. In Gene Co-expression Networks (GCNs) genes correspond to nodes and edges indicate significant correlations among gene expression levels. These undirected graphs reflect coordinated expression patterns and can vary across conditions such as tissue types or disease states, revealing tissue-specific or context-specific gene modules.Conversely, GRNs are typically modeled as directed graphs, where nodes represent genes and edges indicate regulatory influences, often mediated by transcription factors binding to promoter regions or enhancers. These interactions may be inferred through experimental techniques (e.g., ChIP-seq) or computational methods (e.g., mutual information, Bayesian inference) [12]. Edges are often annotated to reflect activation or repression (e.g., +1/−1), enabling GRNs to represent causal or mechanistic regulatory relationships. 

R: Lines 366 and 385 – section titles should not be in Italic font. 

A: Modified in the text. 

R: Line 368: CVAE abbreviation should be given in full (only VAE abbreviation was shown). 

A: Modified in the text  

Line 381 – why ‘glucose concentration’ is in bold font, and next parameters are in Italic font? 

A: Corrected in the text 

The Tables tittles should be written without abbreviations. 

Table 2 – write ‘PCs’ in full. 

Table 3 – write ‘TSTR’ in full 

A: Modified in the text 

R: The references 4,5, 19 are still not formatted (The titles are mixed with the authors names initials) 

A: Modified in the text. 

Reviewer 2 Report

Comments and Suggestions for Authors

All the changes have been incorporated.

Author Response

We thank again the reviewer for their important feedback and comments that helped us improving the quality of the manuscript.

Reviewer 3 Report

Comments and Suggestions for Authors

First of all I want to comment on the author's reply. The example you gave is irrelevant because of two things: a) the deviation of 0.2 is extremely high, in particular that means that 27% of samples become noise values from -0.4..-02, and 0.2… 0.4, so the difference reaches 0.8 !! In that case it is obvious that the correlation would be low. b) you compare the correlation to some absolute values (like 30%), but  you should show the difference in behaviour of your method compared to adding noise. By applying your method may the correlation drop to 50% ?

You state that “they construct new samples that are coherent, internally consistent, and statistically representative”, but you never give any measure of that characteristics. Please show in values how much your method is more consistent than other methods. Same applies to “fairness” . 

“thus increasing their a even unlabeled samples can contribute to training the model by helping it learn basic relationships and patterns within the data”  - please show these relationships (if they exist), that could be a good paper on itself. In the present paper nop relationships shown, neither an ability to catch them. 

The major problem of the work is unclear motivation. Any sensitive personal data is anonymized and shared easily. The huge databases of transcriptomic data, DNA sequences are clear proof of that.  Projects like UK -biobank representing hundreds of thousands of individual genomes with genome variation, transcriptome, proteome etc, all at the level of a single individual. 

Authors must clearly present the use of their method, the area where it can be applied and why it is better. Till now I see the authors actually valued their own work with the following sentence “synthetic datasets may not carry immediate biological meaning on their own”.

Author Response

R: First of all I want to comment on the author's reply. The example you gave is irrelevant because of two things: a) the deviation of 0.2 is extremely high, in particular that means that 27% of samples become noise values from -0.4..-02, and 0.2… 0.4, so the difference reaches 0.8 !! In that case it is obvious that the correlation would be low. b) you compare the correlation to some absolute values (like 30%), but  you should show the difference in behaviour of your method compared to adding noise. By applying your method may the correlation drop to 50% ? 

You state that “they construct new samples that are coherent, internally consistent, and statistically representative”, but you never give any measure of that characteristics. Please show in values how much your method is more consistent than other methods. Same applies to “fairness” . 

A: In the original GAN paper, the authors mathematically demonstrate that the model indirectly minimizes the Jensen–Shannon divergence between the distributions of real and generated data. This implies that, under ideal conditions—such as convergence and optimal training—the distributions of real and synthetic datasets become indistinguishable. 

Conversely, it is straightforward to show that the reverse is not true. It is well understood that if we have a variable distributed as N(μ,σ1 ) and we add Gaussian noise from N(0,σ2 ), the resulting distribution becomes N(μ,σ1 +σ2 ), regardless of how small σ2 is. This clearly alters the original distribution. 

Finally, regarding the evaluation metrics, we have demonstrated that our method improves unsupervised Precision, Recall, and Correlation compared to the baselines. 

R: “thus increasing their a even unlabeled samples can contribute to training the model by helping it learn basic relationships and patterns within the data”  - please show these relationships (if they exist), that could be a good paper on itself. In the present paper nop relationships shown, neither an ability to catch them.  

A: It appears that the reviewer has taken a sentence out of context and introduced a criticism that does not accurately reflect our work. We would like to clarify that we have never claimed to infer knowledge from the generated data, as the comment might suggest. Instead, we explicitly stated that the unlabeled samples are used solely to contribute to the training of the GAN model. 

R: The major problem of the work is unclear motivation. Any sensitive personal data is anonymized and shared easily. The huge databases of transcriptomic data, DNA sequences are clear proof of that.  Projects like UK -biobank representing hundreds of thousands of individual genomes with genome variation, transcriptome, proteome etc, all at the level of a single individual.  

Authors must clearly present the use of their method, the area where it can be applied and why it is better. Till now I see the authors actually valued their own work with the following sentence “synthetic datasets may not carry immediate biological meaning on their own”. 

A: Allow us to organize our response into different sections. To support our claims, we reference review articles published in peer-reviewed journals from 2023 to the present. However, we do not intend to include these citations in the manuscript, as our work is a methodological paper rather than a review. 

  • Motivation: The use of synthetic data has emerged as a transformative solution in numerous applications of AI-driven research. Given that modern healthcare research heavily relies on data-intensive solutions, the generation of synthetic data is, at the very least, a topic worth investigating. While we believe that we have already discussed the potential of synthetic data in this domain in depth, we are glad to observe that the same vision is shared by the community, as demonstrated by several recent publications (Giuffrè, 2023), (Chen, 2024), (Breugel, 2024), (Fabuyi, 2024), (Farhadi, 2025), (Ktena, 2024), (Pezoulas, 2024), (Draghi, 2024). Please note that the venues where those papers have been published include Nature Reviews Bioengineering and Nature Medicine, journals that steer and influence research in this field. 

  • Applications: The reviewer may have overlooked the growing importance of synthetic data across various fields related to bioengineering and biomedicine. As a non-exhaustive list, we would like to highlight a few contributions across different subfields: Histopathology imaging (Carrillo-Perez, 2023), oncological clinical data (Christoforou, 2025), multi-omic datasets (Cai, 2024), (Selvarajoo, 2024), connectome (Vellemer, 2025), radiomics (Koetzier, 2024), and dermatology (Luschi, 2025). 
    We believe that this empirically shows that synthetic data may actually have a role in bioengineering. 

  • Privacy: We respectfully disagree with the reviewer’s statement. First, large-scale data repositories are rarely publicly available. For example, accessing the UK Biobank requires submitting a formal application, and even then, access is restricted to only those data fields deemed necessary for the approved research. Access to the complete set of information for any individual is explicitly prohibited. Second, simply having access to a dataset does not imply that all privacy concerns associated with it have been resolved. In recent years, many studies have investigated the Ethical, Legal and Societal aspects of synthetic data in health care:  (Pasculli, 2025), (Susser, 2024), (Wagner, 2024), (Boraschi, 2024), (Hyrup, 2025), (Quian, 2024), (Mendes, 2025), (Abgrall, 2025), (Chen, 2024) 

 REFERENCES:

(Abgrall, 2025) Abgrall G, Monnet X, Arora A. Synthetic Data and Health Privacy. JAMA. 2025 Feb 18;333(7):567-8. 
(Boraschi, 2024) Boraschi D, van der Schaar M, Costa A, Milne R. Governing synthetic data in medical research: the time is now. The Lancet Digital Health. 2025 Apr 1;7(4):e233-4. 
(Breugel, 2024) van Breugel B, Liu T, Oglic D, van der Schaar M. Synthetic data in biomedicine via generative artificial intelligence. Nature Reviews Bioengineering. 2024 Dec;2(12):991-1004. 
(Cai, 2024) Cai Z, Apolinário S, Baião AR, Pacini C, Sousa MD, Vinga S, Reddel RR, Robinson PJ, Garnett MJ, Zhong Q, Gonçalves E. Synthetic augmentation of cancer cell line multi-omic datasets using unsupervised deep learning. Nature communications. 2024 Nov 29;15(1):10390. 
(Carrillo-Perez, 2023) Carrillo-Perez F, Pizurica M, Ozawa MG, Vogel H, West RB, Kong CS, Herrera LJ, Shen J, Gevaert O. Synthetic whole-slide image tile generation with gene expression profile-infused deep generative models. Cell Reports Methods. 2023 Aug 28;3(8). 
(Chen, 2024) Chen Y, Esmaeilzadeh P. Generative AI in medical practice: in-depth exploration of privacy and security challenges. Journal of Medical Internet Research. 2024 Mar 8;26:e53008. 
(Chen, 2024) Chen M, Mei S, Fan J, Wang M. Opportunities and challenges of diffusion models for generative AI. National Science Review. 2024 Dec;11(12):nwae348. 
(Christoforou, 2025) Christoforou AT, Spohn SK, Sprave T, Fechter T, Rühle A, Nicolay NH, Popp I, Grosu AL, Peeken JC, Thieme AH, Stylianopoulos T. A framework to create, evaluate and select synthetic datasets for survival prediction in oncology. Computers in Biology and Medicine. 2025 Jun 1;192:110198. 
(Draghi, 2024) Draghi B, Wang Z, Myles P, Tucker A. Identifying and handling data bias within primary healthcare data using synthetic data generators. Heliyon. 2024 Jan 30;10(2). 
(Fabuyi, 2024) Fabuyi JA. Leveraging Synthetic Data as a Tool to Combat Bias in Artificial Intelligence (AI) Model Training. Journal of Engineering Research and Reports. 2024 Nov 27;26(12):24-46. 
(Farhadi, 2025) Farhadi A, Taheri A. Application of GenAI in Synthetic Data Generation in the Healthcare System. In Application of Generative AI in Healthcare Systems 2025 Feb 26 (pp. 67-89). Cham: Springer Nature Switzerland. 
(Giuffrè, 2023) Giuffrè M, Shung DL. Harnessing the power of synthetic data in healthcare: innovation, application, and privacy. NPJ digital medicine. 2023 Oct 9;6(1):186. 
(Hyrup, 2025) Hyrup T, Lautrup AD, Zimek A, Schneider-Kamp P. A systematic review of privacy-preserving techniques for synthetic tabular health data. Discover Data. 2025 Dec;3(1):1-32. 
(Koetzier, 2024) Koetzier LR, Wu J, Mastrodicasa D, Lutz A, Chung M, Koszek WA, Pratap J, Chaudhari AS, Rajpurkar P, Lungren MP, Willemink MJ. Generating synthetic data for medical imaging. Radiology. 2024 Sep 10;312(3):e232471. 
(Ktena, 2024) Ktena I, Wiles O, Albuquerque I, Rebuffi SA, Tanno R, Roy AG, Azizi S, Belgrave D, Kohli P, Cemgil T, Karthikesalingam A. Generative models improve fairness of medical classifiers under distribution shifts. Nature Medicine. 2024 Apr;30(4):1166-73. 
(Luschi, 2025) Luschi A, Tognetti L, Cartocci A, Cevenini G, Rubegni P, Iadanza E. Advancing synthetic data for dermatology: GAN comparison with multi-metric and expert validation approach. Health and Technology. 2025 Apr 25:1-0. 
(Mendes, 2025) Mendes JM, Barbar A, Refaie M. Synthetic data generation: a privacy-preserving approach to accelerate rare disease research. Frontiers in Digital Health. 2025 Mar 18;7:1563991. 
(Pasculli, 2025) Pasculli G, Virgolin M, Myles P, Vidovszky A, Fisher C, Biasin E, Mourby M, Pappalardo F, D'Amico S, Torchia M, Chebykin A. Synthetic Data in Healthcare and Drug Development: Definitions, Regulatory Frameworks, Issues. CPT: Pharmacometrics & Systems Pharmacology. 2025 Apr 4. 
(Pezoulas, 2024) Pezoulas VC, Zaridis DI, Mylona E, Androutsos C, Apostolidis K, Tachos NS, Fotiadis DI. Synthetic data generation methods in healthcare: A review on open-source tools and methods. Computational and structural biotechnology journal. 2024 Jul 9. 
(Quian, 2024) Qian Z, Callender T, Cebere B, Janes SM, Navani N, van der Schaar M. Synthetic data for privacy-preserving clinical risk prediction. Scientific Reports. 2024 Oct 27;14(1):25676. 
(Selvarajoo, 2024) Selvarajoo K, Maurer-Stroh S. Towards multi-omics synthetic data integration. Briefings in Bioinformatics. 2024 May 1;25(3):bbae213. 
(Susser, 2024) Susser D, Schiff DS, Gerke S, Cabrera LY, Cohen IG, Doerr M, Harrod J, Kostick‐Quenet K, McNealy J, Meyer MN, Price WN. Synthetic Health Data: Real Ethical Promise and Peril. Hastings Center Report. 2024 Sep;54(5):8-13. 
(Vellemer, 2025) Vellmer S, Aydogan DB, Roine T, Cacciola A, Picht T, Fekonja LS. Diffusion MRI GAN synthesizing fibre orientation distribution data using generative adversarial networks. Communications Biology. 2025 Mar 28;8(1):512. 
(Wagner, 2024) Wagner JK, Cabrera LY, Gerke S, Susser D. Synthetic data and ELSI-focused computational checklists—A survey of biomedical professionals’ views. PLOS Digital Health. 2024 Nov 20;3(11):e0000666. 

Round 3

Reviewer 1 Report

Comments and Suggestions for Authors

Thanks for the manuscript update and detailed answer. I have no more critical remarks.

Please check typo: Footnote 1 on page 7

Author Response

R: Please check typo: Footnote 1 on page 7

A: We thank the reviewer. We have corrected this in the manuscript.

Reviewer 3 Report

Comments and Suggestions for Authors

Summarising again: you present a system which can generate artificial data which is very similar to real data. No example of the use is present. And I cannot even imagine how it can be constructively used. All such kinds of manipulations will not increase the information content to the data, therefore will not lead to better detection of biological principles. 

Overall the use and application area of such artificial data is unclear. 

 “A: It appears that the reviewer has taken a sentence out of context, … we have never claimed to infer knowledge from the generated data” - you try to misinterpret me. It was a citation “ training the model by helping it learn basic relationships” - please agree that you have no idea what your model has learned. That is why you can't present these “relationships and patterns” which you learned from ORIGINAL (not generated! ) data. 

Motivation: 

“Given that modern healthcare research heavily relies on data-intensive solutions” - important which data? Not any data, but well described (raw) experimental data. 

I do not see where you “discussed the potential of synthetic data in this domain in depth”, I am constantly asking to give a reasonable example. When others successfully publish in this domain, does not immediately indicate that your work is of the same value. If you think reviewers at  Nature Medicine are less rigorous, try submitting there. 

Applications:

I have not overlooked the importance of synthetic data across various fields. I cannot see the application of your work. I am constantly asking - give an example. 

Privacy: 

In the UK Biobank users pay only for CPUs, and, here you must agree, it is much (!) cheaper than having an own infrastructure for such amounts of data  - petabytes!  I have access to all fields, same for my colleagues, down to any individual. You may not download the raw data - same way as you cannot copy copyrighted material, but actually 1) you do not need that, 2) just do the analysis in the cloud, down to a particular SNP in each person as we do. You need to try it, it is the best database. 

Apart from this, there are hundreds of other databases such as 1000 genomes project, NCBI GEO etc. and there are no privacy problems. “many studies have investigated the Ethical,... “ - explain readers how you contribute to this. Here it is evaluated what you have done, not others. 

Author Response

R: Summarising again: you present a system which can generate artificial data which is very similar to real data. No example of the use is present. And I cannot even imagine how it can be constructively used. All such kinds of manipulations will not increase the information content to the data, therefore will not lead to better detection of biological principles. Overall the use and application area of such artificial data is unclear.

A: We respectfully acknowledge the reviewer’s comment; however, we find the point raised unclear. The manuscript’s introduction already discusses several potential applications of synthetic data in the biomedical field. Furthermore, the paper includes two concrete examples demonstrating the use of synthetic data generated by our model, specifically applied to E. coli and human transcriptomics datasets.

R: “A: It appears that the reviewer has taken a sentence out of context, … we have never claimed to infer knowledge from the generated data” - you try to misinterpret me. It was a citation “ training the model by helping it learn basic relationships” - please agree that you have no idea what your model has learned. That is why you can't present these “relationships and patterns” which you learned from ORIGINAL (not generated! ) data.

A:  As is often the case with complex artificial intelligence models—particularly those with a large number of parameters (in our case, > 100K)—it is inherently difficult to determine what specific patterns or relationships have been learned. This is a well-known limitation in many deep learning approaches.

However, the utility of such models does not rely solely on our ability to interpret their inner working mechanisms. Instead, they are typically assessed based on their performance on downstream tasks using appropriate validation metrics, which we have reported in the manuscript.

Finally, we would like to emphasize that the patterns and relationships are indeed learned on real (original) data, and the synthetic data it generates is intended to reflect the distribution of that real data. This is in line with the primary objectives of synthetic data generation—namely, enhancing data availability while preserving privacy.

R: Motivation:  “Given that modern healthcare research heavily relies on data-intensive solutions” - important which data? Not any data, but well described (raw) experimental data.  I do not see where you “discussed the potential of synthetic data in this domain in depth”, I am constantly asking to give a reasonable example. When others successfully publish in this domain, does not immediately indicate that your work is of the same value. If you think reviewers at  Nature Medicine are less rigorous, try submitting there.

A: We would like to reiterate that we have already provided two concrete examples in the manuscript where the synthetic datasets generated by our model are used in two scenarios. These examples serve to demonstrate the practical utility of synthetic data in the biomedical domain. 

We are sorry if our message was misunderstood, and we regret that it may have come across as a self-assessment of the value of our contribution. We would like to clarify that our references to related work on synthetic data were not intended to evaluate our own work, but rather to highlight the increasing interest and relevance of synthetic data in the research community, as initially raised by the reviewer and reported below. Throughout the revision process, we have engaged with all reviewer comments respectfully and constructively, and have made sincere efforts to address each concern thoughtfully. 

Applications:

R: I have not overlooked the importance of synthetic data across various fields. I cannot see the application of your work. I am constantly asking - give an example. 

A: We respectfully note that, throughout the paper, we have provided two concrete examples demonstrating the usefulness of our approach: (1) training machine learning classifiers on synthetic data to predict disease states, and (2) predicting pH levels in E. coli samples. In both cases, classifiers trained on our synthetic data achieved high performance, confirming that the generated data can be as informative and effective as real data.

Beyond these examples, our work introduces a generative model infused with biological prior knowledge. This has significant implications for the research community: such model can be used to simulate specific biological conditions—including rare or underrepresented scenarios—at virtually no cost. The resulting synthetic samples can augment real datasets, reduce reliance on sensitive patient data, and accelerate research. This ultimately leads to substantial reductions in both time and cost. 

R: Privacy: 

In the UK Biobank users pay only for CPUs, and, here you must agree, it is much (!) cheaper than having an own infrastructure for such amounts of data  - petabytes!  I have access to all fields, same for my colleagues, down to any individual. You may not download the raw data - same way as you cannot copy copyrighted material, but actually 1) you do not need that, 2) just do the analysis in the cloud, down to a particular SNP in each person as we do. You need to try it, it is the best database. 

Apart from this, there are hundreds of other databases such as 1000 genomes project, NCBI GEO etc. and there are no privacy problems. “many studies have investigated the Ethical,... “ - explain readers how you contribute to this. Here it is evaluated what you have done, not others. 

A: We fully agree on the immense value of resources such as the UK Biobank, the 1000 Genomes Project, and NCBI GEO. Platforms like the UK Biobank have indeed significantly lowered the barrier to large-scale biomedical data analysis.

However, we would like to clarify that our contribution is not intended to replace such infrastructures, but rather to complement them. Our generative model introduces a novel way to simulate realistic omics data under customizable biological conditions—including rare or privacy-sensitive scenarios that are often underrepresented or entirely absent in real-world datasets. Even under optimal conditions of data access, certain pathological or combinatorial cases may simply not occur frequently enough to be captured. Our approach enables the simulation of such cases at no additional experimental cost. While our paper presents two practical applications—disease classification and pH prediction in E. coli—our broader goal is to provide a general-purpose tool that can support such tasks across a wide range of biological contexts.

From an ethical standpoint, our work goes beyond referencing existing studies. We present a concrete generative model that integrates biological prior knowledge to produce plausible omics samples. These are rigorously validated both in terms of biological realism and performance in downstream machine learning tasks—using the same evaluation criteria commonly adopted in the literature for such models. This ensures that our approach can support real-world biomedical research while minimizing the risks of data leakage or re-identification.

We hope this clarifies the scope and impact of our contribution.